# Chameau (HBO1) regulates starvation resistance in *Drosophila melanogaster* in a temperature-dependent manner

Anuroop Venkateswaran Venkatasubramani[1,2], Toshiharu Ichinose[3,4], Ignasi Forne[5], Nathaniel W Snyder[6], Hiromu Tanimoto[3], Shahaf Peleg[7], Axel Imhof[1,5]

The body temperature of *Drosophila melanogaster* depends on the extrinsic temperature. Numerous studies show that environmental temperature influences metabolism, lifespan, and starvation resilience. We have previously shown that Chameau (Chm), a MYST-domain acetyltransferase, promotes aging but also increases starvation resilience. Strikingly, the metabolic increase associated with a 2°C temperature rise was sufficient to bypass the requirement for Chm in starvation resilience, suggesting that Chm modulates metabolism in a temperature-dependent manner. The increase in temperature also rescued the dampened expression of genes involved in starvation response, the weight loss, and the misregulation of trehalose, which we observed in *chm* mutants at 23°C. Thus, Chm regulates starvation at ≤23°C but becomes obsolete at higher temperatures, likely because of efficient acetyl-CoA generation ensuring similar acetylation despite lower Chm. Supporting this, citrate supplementation increased starvation resilience of *chm* mutants at lower temperatures. Our finding that a gene's role manifests only under specific environments has important implications in light of global climate change.

## Introduction

One of the major environmental factors that shapes the behavior of the organisms is temperature. The change in temperature affects many physiological processes such as basal metabolic rate, development, longevity, and survival, especially in cold-blooded organisms (Pijpe et al, 2007). Insects, fish, and nematodes are poikilothermic organisms whose body temperature changes with the environmental conditions and therefore have developed various adaptations to react to alterations in the environment (Fast et al, 2017; Mołoń et al, 2020). In a constantly changing environment, adaptation to such novel conditions becomes key for survival. Therefore, poikilothermic organisms can change their behavior or physiology in response to even very subtle environment changes (Karan & David, 2000; Pijpe et al, 2007; Padmanabha et al, 2011; Jang & Lee, 2018).

One such organism that has populated most habitats of the world is the fruit fly, *Drosophila melanogaster*. Multiple studies have shown the influence of temperature on its development, lifespan, metabolism, and physiology. For example, an increase in temperature accelerates development, metabolism, and aging in fruit flies (Mołoń et al, 2020; Goh et al, 2021; Lin et al, 2023). Even though evidence suggests that subtle changes, such as an increase of 2–3°C, can have a substantial effect on fly physiology (Mołoń et al, 2020), most studies that address the molecular pathways involved in adaptation focus on relatively large changes in conditions, such as heat or cold shock.

We have recently shown that reduction of the acetyltransferase Chameau (Chm) results in an impaired starvation response when measured at 23°C. We have also detected that mutants of *chm* display substantial changes in the expression of metabolic genes, proteins, and the posttranslational modification (PTMs) of proteins affecting physiology and starvation resilience (Venkatasubramani et al, 2023). Interestingly, population genetic studies in *Drosophila* identified Chm as one of the chromatin factors that are differentiated between tropical and temperate populations of *D. melanogaster*. The *chm* gene region contains single nucleotide polymorphisms (SNPs) in varying densities depending on the ambient conditions of the fly population (Levine & Begun, 2008; Croze et al, 2017). Accordingly, we were wondering whether the role of Chm in mediating starvation response is sensitive to differences in environmental temperatures. Notably, our laboratory has

[1]Department of Molecular Biology, Biomedical Center Munich, Faculty of Medicine, LMU Munich, Martinsried, Germany [2]Graduate School of Quantitative Biosciences (QBM), LMU Munich, Munich, Germany [3]Graduate School of Life Sciences, Tohoku University, Sendai, Japan [4]Frontier Research Institute for Interdisciplinary Sciences, Tohoku University, Sendai, Japan [5]Protein Analysis Unit, Biomedical Center Munich, Faculty of Medicine, LMU Munich, Martinsried, Germany [6]Aging and Cardiovascular Discovery Center, Lewis Katz School of Medicine, Temple University, Philadelphia, PA, USA [7]Research Group Epigenetics, Metabolism and Longevity, Institute for Farm Animal Biology, Dummerstorf, Germany

Correspondence: peleg@fbn-dummerstorf.de; imhof@lmu.de
Anuroop Venkateswaran Venkatasubramani's present address is Max-Delbrück-Centrum für Molekulare Medizin (MDC), Berlin, Deutschland

already shown that exposure of fruit flies to different temperatures results in altered activity of enzymes involved in citrate synthesis and acetyl-CoA and correspondingly changes in those metabolites and downstream histone acetylation (Peleg et al, 2016). These findings raise the possibility that Chm may play a selective role in the adaptation and survival upon temperature change.

To explore this, we decided to assess the effects of temperature on *chm* mutants and its ability to survive starvation. Our data show that the starvation phenotype of *chm* mutants was indeed temperature-specific. We also observed an effect of temperature on the molecular changes observed in *chm* mutant fly strains such as the changes in histone PTMs, the transcriptome, and the metabolome. Our observations confirm the role of Chm as an enzyme that mediates adaptation to stress under very specific environmental conditions.

# Results

### Mild temperature changes affect starvation response in Chameau mutant flies

To test the influence of temperature on starvation resilience, we raised flies within a temperature range from 18°C to 27°C. As previously shown, $chm^{MYST/+}$ flies show impaired survival upon starvation at 23°C (Venkatasubramani et al, 2023), which was even larger at lower temperatures. However, at temperatures of 25°C and above, the phenotype was no longer observed (Figs 1A and B and S1A–D). Although male and female fruit flies have differences in starvation susceptibility (Chandegra et al, 2017; Lin et al, 2023), they show a similar temperature dependency (Figs 1C and D and S1E–H). This was independent of geographic location as similar results were observed from two different laboratories in Germany and Japan (Fig S2A and B; see Fly maintenance and Starvation assay sections in the Materials and Methods section). We also observed a similar phenotype in a *chm* RNAi line ($chm^{RNAi}$, crossed with *da-Gal4*) in both males and females, further validating temperature-dependent differences in starvation resilience of flies with a reduced *chm* expression (Figs 1E and S1I and J). Taken together, we concluded that the impaired starvation in *chm* mutants compared with control flies is temperature-dependent and is manifested only at colder temperatures.

As fruit flies have been shown to consume food differently when grown at different temperatures (Klepsatel et al, 2019), we measured the food consumption in control and $chm^{MYST/+}$ male flies at temperatures of 23°C and 25°C. Food intake and excretion showed no differences at 23°C and a marginal reduction in $chm^{MYST/+}$ flies at 25°C, which was not statistically significant (Fig S3A and B). Overall, the total food consumption showed negligible differences irrespective of both genotype and temperature (Fig S3C).

### Profiling of histone posttranslational modifications indicates the role of acetylation in genotype, temperature, and starvation response

Chm is an acetyltransferase that has been shown to affect H3 and H4 acetylation (Feller et al, 2015; Nakagawa et al, 2015). Therefore,

we compared the histone modification levels of control and $chm^{MYST/+}$ male flies maintained at 23°C and 25°C under fed and starved conditions (Table S1). A principal component analysis of H4 acetylation modification patterns revealed a separation based on both temperature (PC1) and genotype (PC2) (Fig 2A). The modifications that contributed the most to the separation based on the genotype (PC2) was H4K12ac, which is consistent with our previous finding that Chm is the major H4K12 acetyltransferase in *Drosophila* (Feller et al, 2015). The first principal component (PC1, temperature) in contrast is dominated by the acetylation of H4K16 and H4 polyacetylations (Fig 2B). To investigate the change in histone acetylation in response to starvation, we measured the acetylation levels after 10 h (early) or 48 h (late) of starvation. As expected, *chm* mutants had reduced H4K12ac irrespective of the temperature, which further decreased upon starvation at 23°C but not at 25°C (Fig 2C and D). The principal component analysis (PCA) of H3 acetylation showed a distinct clustering based only on temperature (PC1) but not on genotype, which was dominated by H3K23ac and H3K18acK23ac (Figs 2E–H and S4A). These results suggest that a more active metabolism at 25°C might lead to increased levels of acetyl-CoA, which in turn results in higher acetylation levels making histone acetylation an important metabolic sensor in flies (Charidemou & Kirmizis, 2024). In contrast to the changes in acetylation, we did not detect changes in H3 methylation at different temperatures or in $chm^{MYST/+}$ flies or during starvation (Fig S4B). However, starvation may still result in changes of histone methylation at distinct loci and therefore also contribute to metabolic sensing.

Collectively, our detailed profiling of histone PTMs indicates that a temperature increase of 2°C is sufficient to at least partially rescue the impaired histone acetylation in fruit flies lacking the acetyltransferase Chm.

### Transcriptomic effects of *chm* are temperature-dependent

To assess whether a change in *chm*'s expression with temperature or during starvation could be the cause of the observed difference in starvation resilience, we measured *chm* mRNA levels in control and *chm* mutant flies at 23°C and 25°C, before and after 24 h of starvation. However, *chm* mRNA levels remained unchanged across conditions (Fig S5).

We next performed a global transcriptomic analysis on fly heads at 25°C to compare the different transcriptional response at 25°C with our previous data from 23°C (Fig S6A; Venkatasubramani et al [2023]). We decided to use fly heads as we have previously shown that loss of *chm* in both neurons and fat body results in reduced starvation resilience (Venkatasubramani et al, 2023), and Chm is strongly expressed in fat body and neurons (Jenkins et al, 2022), both of which are present in the fly head (Tatar et al, 2014). Similar to what we observed at 23°C, a large percentage (~60%) of transcriptional variance among all samples was accounted for by changes in the nutrient status (starvation), whereas only 10% could be explained by the different genotypes (Fig S6B; Venkatasubramani et al [2023]). Consistently, relatively few genes were differentially expressed between genotypes under either fed (298 genes) or starved (193 genes) conditions, with only limited overlap (65 genes) (Fig S7A; Tables S2 and S3).

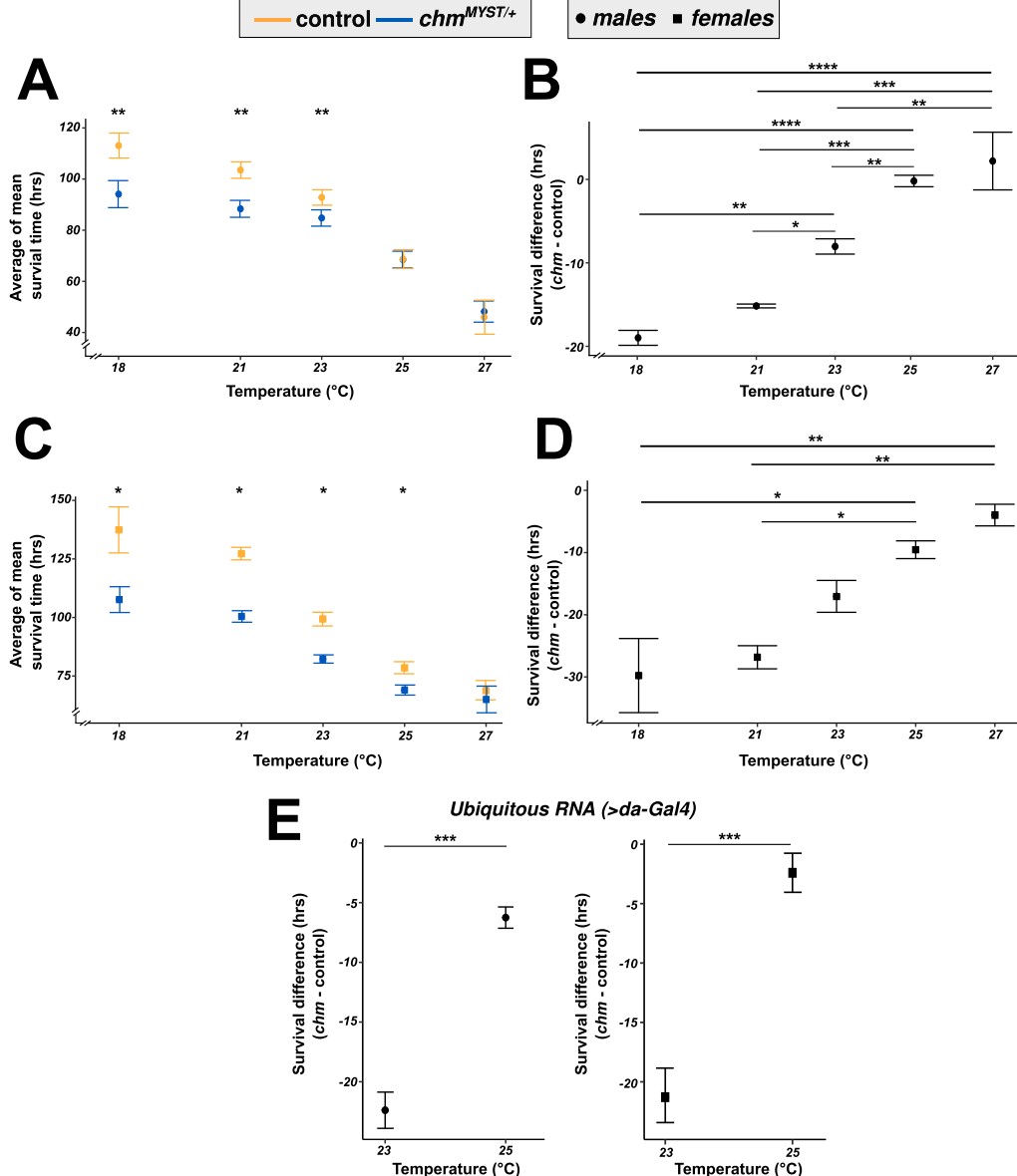

**Figure 1. Temperature changes affect the starvation response in *chm* mutant flies irrespective of the gender.**

**(A, B, C, D)** Average mean survival time (A) and survival difference (B) in males and females (C, D) of control and *chm^{MYST/+}* mutants upon starvation at temperatures of 18°C (N = 3, paired), 21°C (N = 3, paired), 23°C (paired data obtained from Venkatasubramani et al [2023]), 25°C (N = 5 [males] and 4 [females], paired), and 27°C (N = 3, paired). Survival difference (difference in mean survival time in hours between mutant and control) in male (left) and female (right) flies at 23°C (paired data obtained from Venkatasubramani et al [2023]) and 25°C (N = 4, paired) in *chm^{RNAi}* flies as compared to its corresponding control. **(E)** Nonsignificant values are not shown (\*P < 0.05, \*\*P < 0.01, \*\*\*P < 0.001, \*\*\*\*P < 0.0001). **(A, B, C, D, E)** Each replicate had at least 100 flies, and a paired *t* test with FDR correction was performed for data in (A, C, E), whereas ANOVA followed by Tukey's test correction was performed for data in (B, D). Error bars represent the SEM.

Source data are available for this figure.

Given the modest genotype effect, we focused on the starvation response. At 25°C, 1,090 and 978 genes were significantly regulated by starvation in control and *chm* mutants, respectively, of which ~700 were shared (Fig S7B; Tables S4 and S5). GO enrichment analysis revealed similar functional categories between genotypes and across temperatures (Fig S6C and D; Tables S6 and S7 Venkatasubramani et al [2023]). However, starvation-induced expression changes in control and *chm* mutants correlated poorly between 23°C and 25°C (Spearman's r = 0.30–0.34) (Fig 3A), confirming previous experiments that the flies respond very differently to starvation depending on the surrounding temperature (Jang & Lee, 2018; Klepsatel et al, 2019).

Comparison of starvation effects across temperatures in wild-type and *chm^{MYST/+}* flies revealed a similar degree of shared and temperature-specific differentially expressed genes (Fig 3B). Over-representation analysis of shared genes indicated enrichment in various metabolic pathways (Fig S8A and B; Tables S8 and S9). Temperature-specific effects were also evident: at 25°C, controls showed enrichment in carbohydrate metabolism genes, whereas at 23°C, mitochondrial genes were more frequently affected (Fig 3C; Tables S10 and S11). In *chm* mutants, starvation at 25°C specifically altered insulin, hormone, and starvation-responsive genes, whereas it primarily affected broader metabolic processes at 23°C (Fig 3D; Tables S12 and S13).

Insulin and hormonal pathways are known to mediate temperature sensing and feeding behavior (Enell et al, 2010; Koyama et al, 2020). We therefore performed an unsupervised hierarchical clustering analysis of the expression of all genes detected within these sets under fed or starved conditions. At both temperatures, two main clusters emerged, primarily defined by the nutritional

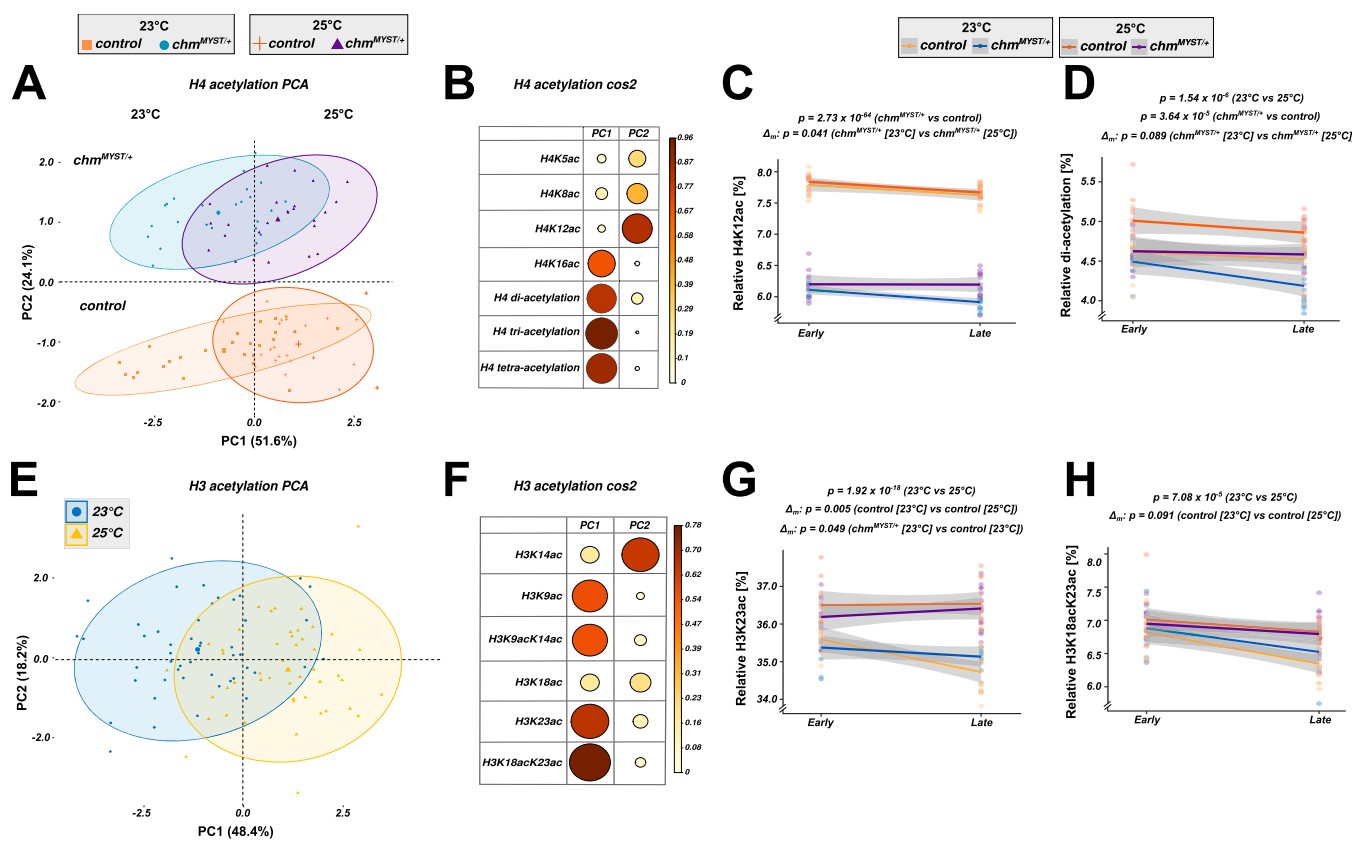

**Figure 2. Histone posttranslational modification profiling indicates the temperature- and starvation-specific changes in H3 and H4 acetylation.**
**(A)** Principal component analysis (PCA) of H4 acetylation data from mass spectrometry between control and $chm^{MYST/+}$ male mutants at 23°C and 25°C in early (0–10 h) and late (24–48 h) time points (N = 5, unpaired) (A). **(A, B)** Cos2 plot of (A) with PC1 and PC2 contributions for H4 acetylation. **(D)** H4K12ac (C) and H4 diacetylation (D) dynamics between control and $chm^{MYST/+}$ male mutants at 23°C and 25°C in early (0–10 h) and late (24–48 h) time points (N = 5). **(E)** PCA of H3 acetylation data from mass spectrometry between control and $chm^{MYST/+}$ male mutants at 23°C and 25°C in early (0–10 h) and late (24–48 h) time points (N = 5, unpaired). **(E, F)** Cos2 plot of (E) with PC1 and PC2 contributions for H3 acetylation. **(G, H)** H3K23ac and (H) H3K18acK23ac dynamics between control and $chm^{MYST/+}$ male mutants at 23°C and 25°C in early (0–10 h) and late (24–48 h) time points (N = 5, unpaired). Nonsignificant values are not shown, and significant $P$-values are mentioned in the figure. All statistical tests were obtained from a linear regression model between relative percentage and condition with time as an interaction term, that is, variable ~ condition ([genotype_temperature] * Time_group) ($\Delta_m$ indicates the change or difference in slope of the mentioned factors). Ellipses for PCA were drawn unsupervised using assuming multivariate t-distribution and bootstrapped statistics (check fviz_pca_ind() or ggplot2::stat_ellipse() function).

state of the flies. In line with the reduced starvation resilience and altered histone acetylation observed in *chm* mutants at 23°C, the starved cluster at this temperature contained almost exclusively control flies, whereas clustering was independent of genotype at 25°C (Fig 3E). This stronger genotype-specific transcriptional response at lower temperatures further supports the hypothesis that Chm acts as a selective metabolic modulator. Based on these data, we next investigated whether temperature-dependent transcriptional changes are reflected in the composition of selected metabolites and the physiological response of the flies to starvation.

### Chm regulates weight and trehalose homeostasis in a temperature-dependent manner

Mutant c$hm^{MYST/+}$ flies show reduced body weight under fed conditions at 23°C, but not at 25°C (Fig 4A; Venkatasubramani et al [2023]). Upon starvation, both genotypes lost weight similarly at 25°C, whereas control flies lost significantly more weight than *chm* mutants at 23°C (Fig 4B; Table S14), further supporting the role of

Chm in temperature-specific metabolic regulation during starvation.

To explore this further, we measured key metabolites such as glucose, glycogen, triacylglycerides (TAGs), and trehalose under fed and starved conditions at both temperatures. Glucose was consistently lower in *chm* mutants regardless of temperature (Fig 4C), possibly contributing to their enhanced lifespan (Venkatasubramani et al, 2023). Starvation led to reduced glucose levels across genotypes (Fig 4D; Table S14), with a faster drop at 23°C compared with 25°C. TAGs and glycogen showed no differences at fed conditions irrespective of genotype or temperature (Fig S9A and C). As expected, both were reduced upon starvation without significant genotype or temperature interaction (Fig S9B and D; Table S14).

In contrast, trehalose regulation was strongly temperature- and genotype-dependent. At 23°C, *chm* mutants had elevated basal trehalose levels but failed to further accumulate trehalose during starvation, unlike controls which showed robust trehalose induction at both early and late time points of starvation

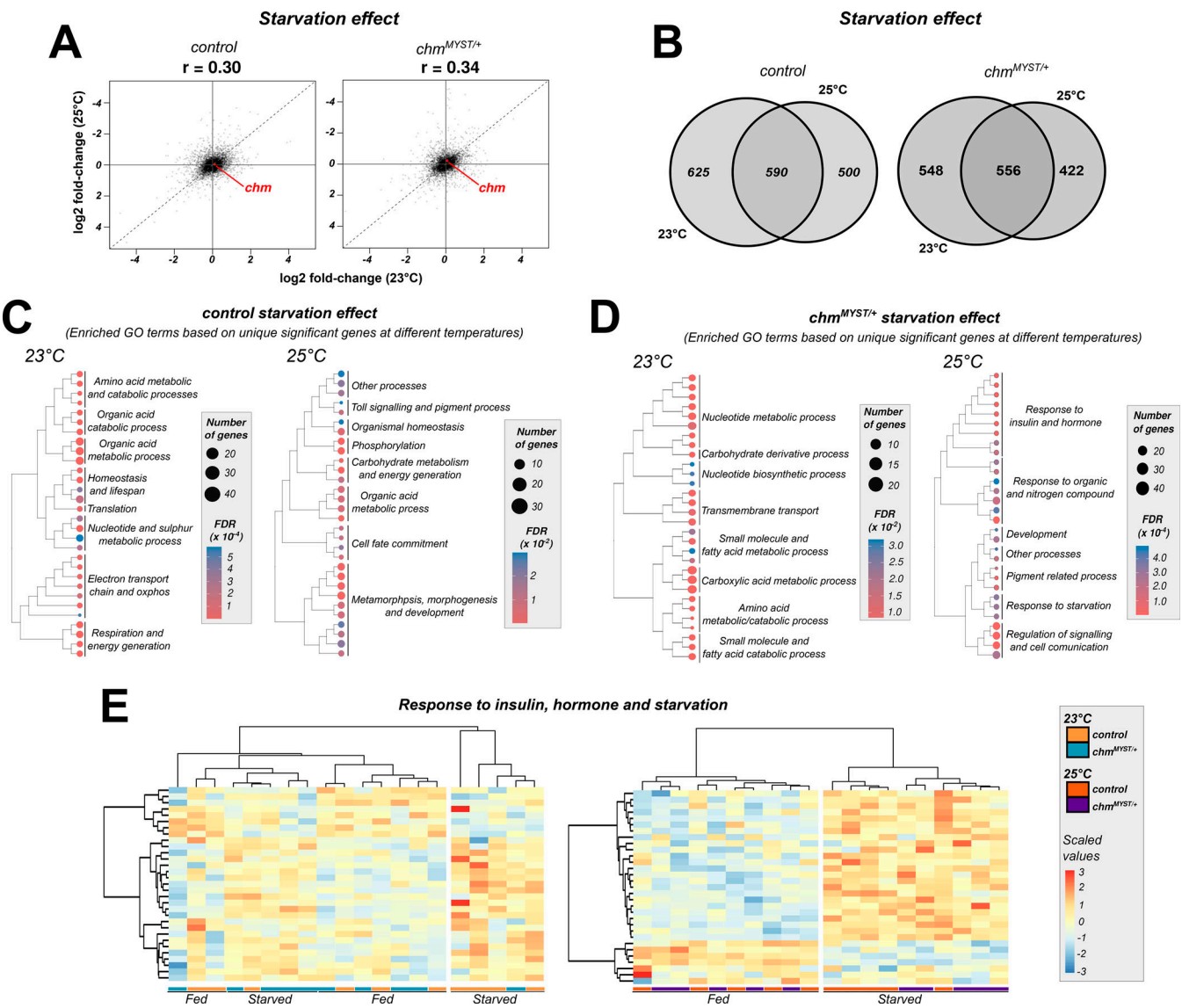

**Figure 3. Transcriptomic data show temperature- and genotypic-specific differences in genes related to insulin and hormonal response.**
**(A)** $\log_2$FC-$\log_2$FC correlation plot of control and $chm^{MYST/+}$ starvation at 23°C (data obtained from Venkatasubramani et al [2023]) and 25°C (N = 5, unpaired). **(B)** Venn diagram showing the number of unique and shared significant genes (FDR and Π-value cutoff ≤ 0.05; refer Materials and Methods section for Π-value) upon starvation in control and $chm^{MYST/+}$ starvation at 23°C and 25°C. **(C, D)** Treeplot showing the top 30 significant GO terms (FDR ≤ 0.05) from over-representation analysis of unique significant genes in control and $chm^{MYST/+}$ starvation at 23°C (left) and 25°C (right), respectively (GO terms were clustered based on semantic similarity, and the terms that were represented the most within a cluster were mentioned). **(E)** Heatmap of scaled read counts of annotated genes from insulin, starvation, and hormonal response in control and $chm^{MYST/+}$ starvation at 23°C (left) and 25°C (right).

(Fig 4E and F; Table S14). At 25°C, both genotypes responded similarly, with a moderate trehalose increase upon starvation. In light of this observation, we checked the mRNA levels of trehalose metabolism-related genes and found minimal differential regulation between both genotypes (control and $chm^{MYST/+}$) and temperatures (Fig S10). However, our proteomics and acetylome analyses of control and $chm^{RNAi}$ flies at 23°C revealed a significant reduction in protein abundance, along with a notable loss of acetylation (Figs S11 and S12). These patterns suggest that Chm is required for proper trehalose mobilization in response to starvation, specifically at lower temperatures.

The ectopic expression of $chm$ in the $chm^{MYST/+}$ background at 23°C resulted in similar weight loss as in wild-type flies. The increase in Chm levels also led to lower trehalose levels that increased upon starvation, similar to what we observed in wild type, validating Chm's role in regulating these responses (Fig 5A–C; Table S15). Consistent with the genotype-independent changes of glucose upon starvation, ectopic $chm$ expression had no significant effect on glucose levels (Fig 5D and E; Table S15). In summary, these results highlight and validate the role of Chm in regulating weight and trehalose levels at lower temperatures. This supports our conclusion that at lower temperatures, starvation resilience

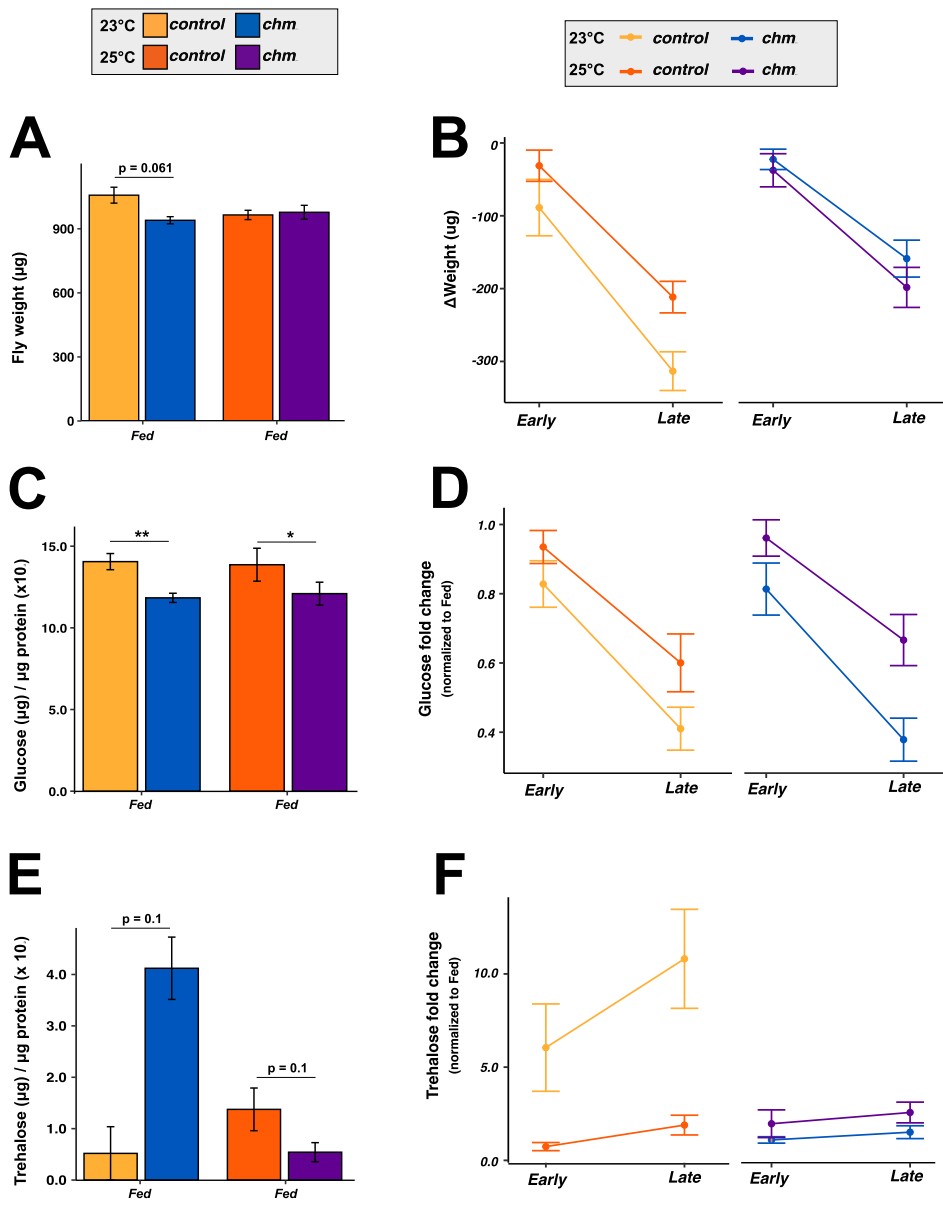

**Figure 4.  *chm* mutant flies showed impaired weight and trehalose homeostasis upon starvation at 23°C.**
**(A, B, C, D, E, F)** Weight (per fly) and corresponding interaction plot (B, C) glucose (µg) (per µg protein) and corresponding interaction plot (D, E) trehalose (µg) (per µg protein) and corresponding interaction plot (F) in control and *chm*^MYST/+ flies at 23°C and 25°C under ad libitum conditions, respectively (N = 4 and 3 [for 23°C trehalose], paired, at least 30 flies per replicate). **(A, C, E)** Paired *t* test with FDR correction was performed for (A, C, E), and only significant values are mentioned. **(B, D, F)** Statistical analysis of (B, D, F) was performed using three-way ANOVA by considering all possible interaction terms, followed by *P*-value adjustment based on interaction terms, wherever necessary. Formula used: Dependent variable ~ Genotype * Temperature * Time_group + Error (Batch). Significant values for all interaction terms are provided in Table S14. Error bars represent the SEM.
Source data are available for this figure.

critically depends on Chm dosage. In contrast, at higher temperatures, increased metabolic activity likely compensates for reduced Chm levels. These findings reinforce our model that Chm integrates metabolic signals to regulate gene expression and physiology. Based on this, we hypothesize that boosting key metabolites, such as acetate or citrate, could similarly bypass the requirement for full Chm function under nutrient stress at low temperatures.

## Citrate supplementation improves starvation survival in *chm* mutants at low temperature

To explore temperature- and genotype-specific metabolic differences, we profiled 32 metabolites from fly heads of fed and starved *chm*^MYST/+ and control flies at 23°C and 25°C (Table S16). Among these,

pyruvate, lactate, and citrate, three interconnected metabolites that feed into acetyl-CoA synthesis, showed notable trends. Although pyruvate and lactate displayed a starvation-dependent increase in *chm*^MYST/+ flies at 23°C, citrate levels were strongly induced upon starvation in *chm*^MYST/+ flies at 25°C (Fig 6A–C; Table S17).

We also measured acetyl-CoA, a key metabolite linking nutrient breakdown to epigenetic regulation and energy metabolism. Although some genotype–time interactions were observed, results were inconsistent across replicates, likely because of acetyl-CoA's instability (Fig 6D; Table S17).

To test whether the increase in citrate at 25°C contributes to improved survival of *chm* mutants, we supplemented sodium citrate during starvation at 23°C. Strikingly, citrate significantly improved starvation resistance in *chm*^MYST/+ flies, restoring

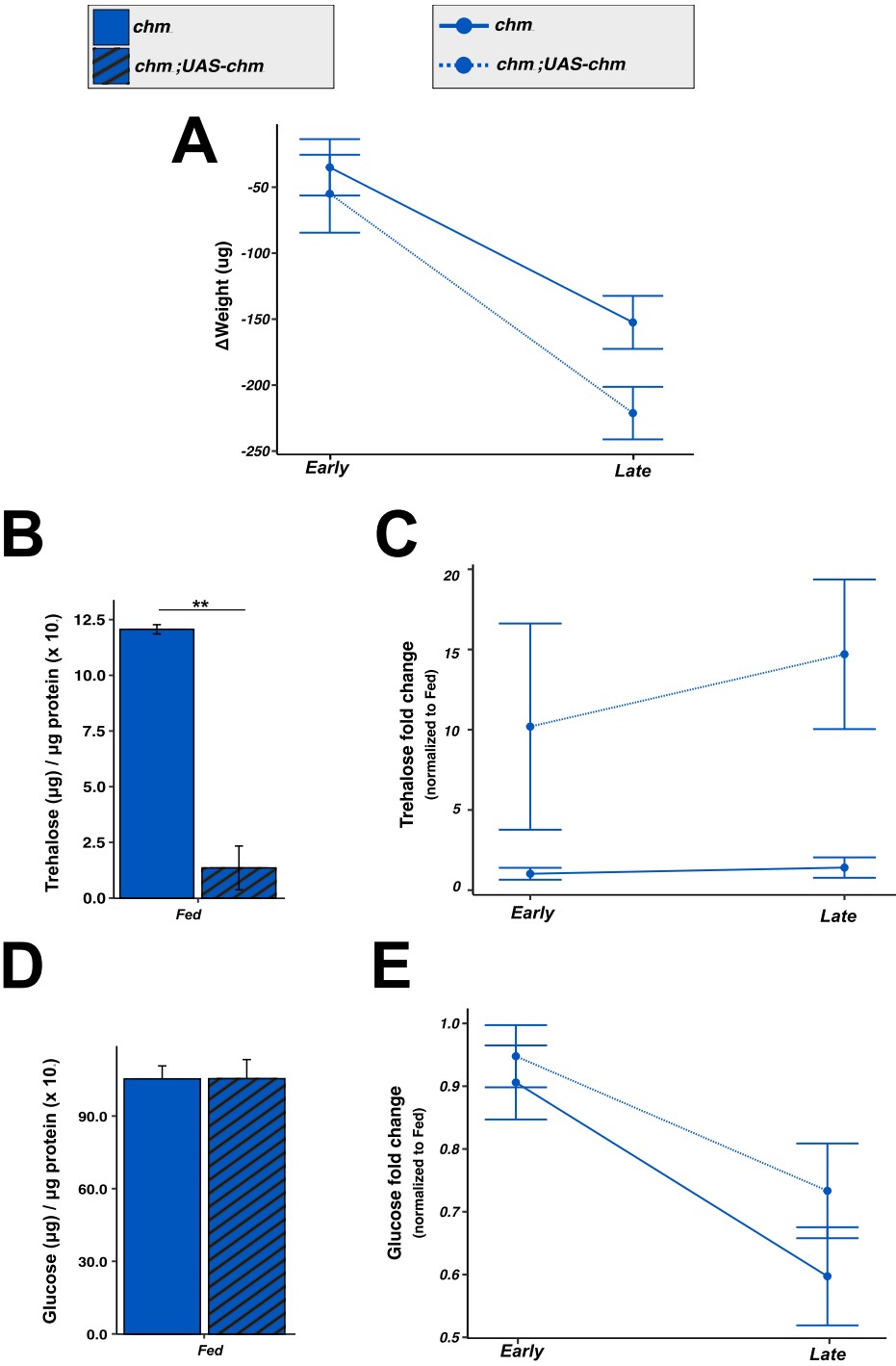

**Figure 5. Weight and trehalose homeostasis is improved upon *chm* overexpression at 23°C.** **(A, B, C, D, E)** Interaction plot depicting change in weight, (B) trehalose ($\mu g$) (per $\mu g$ protein) and the corresponding interaction plot (C), and (D) glucose ($\mu g$) (per $\mu g$ protein) and the corresponding interaction plot (E) upon starvation in $chm^{MYST/+}$ and $chm^{MYST/+};UAS$-$chm^{myc}$ flies at 23°C. All values are normalized to the fed condition of corresponding genotype and temperature (N = 4 and 3 [for trehalose], paired, at least 30 flies per replicate). **(B, D)** Paired *t* test was performed for (B, D), and only significant values are mentioned. **(A, C, E)** Statistical analysis of (A, C, E) was performed using two-way ANOVA by considering all possible interaction terms, followed by *P*-value adjustment based on interaction terms, wherever necessary. Formula used: Dependent variable ~ Genotype * Time_group + Error (Batch). Significant values for all interaction terms are provided in Table S15. Error bars represent the SEM.
Source data are available for this figure.

survival to near control levels, while only marginally improving survival in controls (Fig 6E; Table S18). Acetate supplementation also increased survival but did so across genotypes, indicating that the effect is not genotype-specific (Fig 6F; Table S18).

Together, these results suggest that citrate metabolism can compensate for reduced Chm activity during starvation, specifically at higher temperatures. This supports a model where an elevated metabolic flux at 25°C bypasses the need for full Chm function, and where metabolite supplementation, particularly citrate, can restore starvation resilience at lower temperatures (Fig 6G).

## Discussion

Our study reveals a critical and temperature-sensitive role of the MYST-family histone acetyltransferase Chm in maintaining

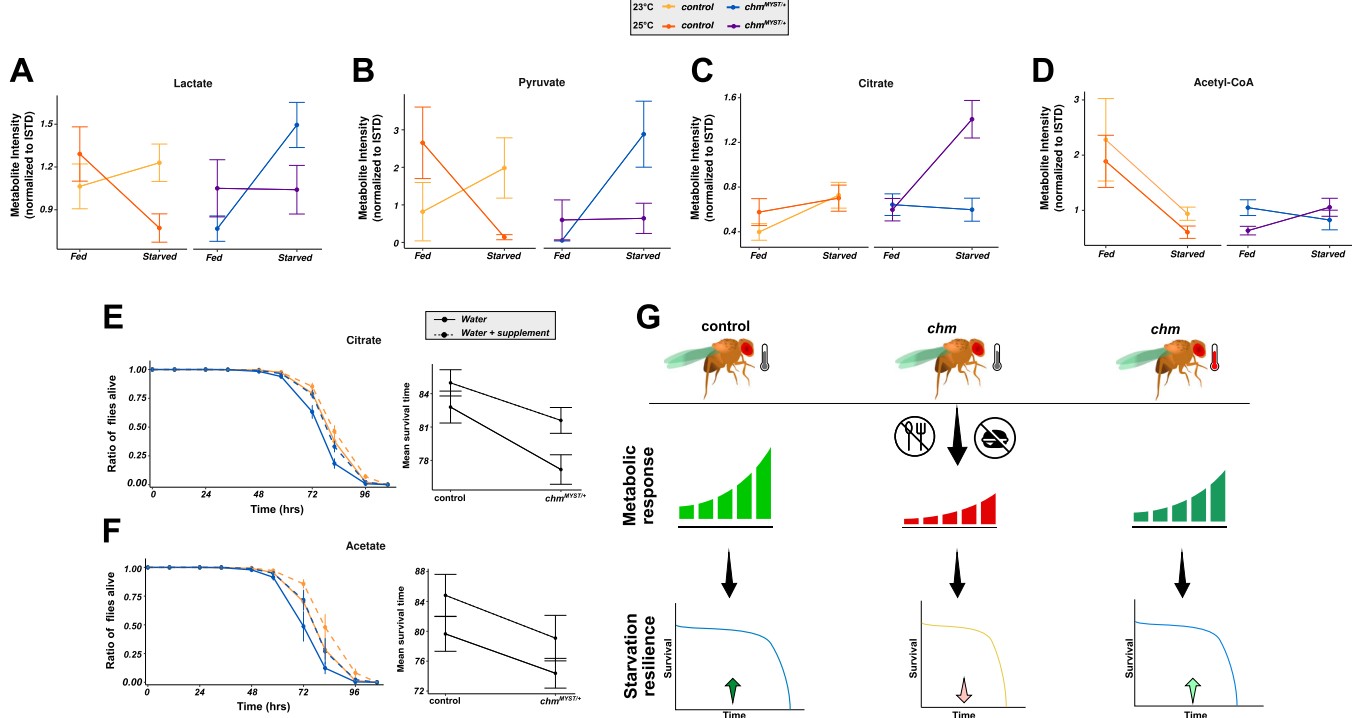

**Figure 6. *chm* mutants show increased citrate levels upon starvation specifically at 25°C.**
**(A, B, C, D)** Interaction plots depicting (A) lactate, (B) pyruvate, (C) citrate, and (D) acetyl-CoA change upon starvation in control and *chm*^MYST/+^ flies at 23°C and 25°C (N = 5, unpaired, ~100 fly heads per replicate). **(E)** Kaplan–Meier plot (left) of control and *chm*^MYST/+^ flies at 23°C with or without acetate supplementation and the corresponding interaction plot (right) of mean survival time (N = 4, paired, at least 100 flies per replicate). **(F)** Kaplan–Meier plot (left) of control and chm^MYST/+^ flies at 23°C with or without citrate supplementation and the corresponding interaction plot (right) of mean survival time (N = 6, paired, at least 100 flies per replicate). **(G)** Model depicting that temperature changes modulate metabolic response, which modifies the starvation response of *chm*^MYST/+^ flies. **(A, B, C, D)** Statistical analysis of (A, B, C, D) was performed using three-way ANOVA by considering all possible interaction terms, followed by *P*-value adjustment based on interaction terms, wherever necessary. Formula used: Dependent variable ~ Genotype * Temperature * Time_group. Significant values for all interaction terms are provided in Table S17. **(E, F)** Statistical analysis of interaction plots in (E, F) was performed using two-way ANOVA by considering all possible interaction terms, followed by *P*-value adjustment based on interaction terms, wherever necessary. Formula used: Dependent variable ~ Genotype * Supplementation. Significant values for all interaction terms are provided in Table S18. Error bars represent the SEM.
Source data are available for this figure.

starvation resilience in *Drosophila*. Strikingly, *chm* mutants exhibit reduced starvation resistance only at temperatures below 23°C, whereas this phenotype is abolished at higher temperatures. This observation aligns with previous studies demonstrating that metabolic rates in poikilothermic organisms, including *Drosophila*, are highly temperature-dependent, with elevated temperatures accelerating metabolic processes (Karan & David, 2000; Mołoń et al, 2020; Goh et al, 2021). At higher temperatures (≥25°C), increased metabolic flux likely compensates for the loss of *chm* by potentially providing sufficient acetyl-CoA to maintain global protein and histone acetylation levels (Padmanabha et al, 2011), thereby sustaining starvation-responsive gene expression. In contrast, at lower temperatures, where metabolic activity is reduced, *chm* might become essential to maintain acetylation capacity, thereby enabling an effective starvation response. Such a critical role of *chm* at 23°C in the starvation response is also supported by our observation that the level of acetylation drops faster in *chm* mutants at 23°C than in *chm* mutants at 25°C during starvation and that we observe elevated histone polyacetylation levels in flies reared at 25°C compared with those at 23°C.

Our observation that supplementation with citrate (a precursor of acetyl-CoA) rescues the starvation sensitivity phenotype of chm mutants at lower temperatures supports the hypothesis that levels of acetyl-CoA serve as modulators of a starvation response. However, acetate supplementation improved survival upon starvation independently of genotype, suggesting a complex network of metabolic processes and starvation responses that extend beyond the action of Chm.

## Developmental compensation and metabolic memory

Beyond its acute role in adult starvation resistance, *chm* appears to function as a compensatory factor for metabolic fluctuations during development. Our data demonstrate that *chm* mutant flies exposed to low temperatures during larval stages exhibit impaired starvation resilience as adults (Fig S13A and B), mirroring findings by Jang and colleagues (Jang & Lee, 2018), who reported that larvae raised at cooler temperatures emerge as adults with reduced starvation resistance. This suggests that *chm* helps buffering

against metabolic limitations imposed by colder developmental conditions, ensuring that adult flies retain the capacity to mobilize energy reserves during periods of scarcity.

Notably, *chm* mutants already display elevated circulating trehalose levels and reduced body weight at 23°C even under fed conditions. This metabolic dysregulation implies that when Chm levels are reduced, flies fail to properly suppress catabolic processes or carbohydrate mobilization, leading to premature depletion of energy stores. Such a phenotype is consistent with a model in which *chm* maintains metabolic homeostasis by modulating chromatin states that regulate energy storage and use. Under low-temperature conditions, where acetyl-CoA is limiting, *chm* may be particularly crucial for down-regulating genes or inactivating proteins that would otherwise promote wasteful energy expenditure, thereby preserving starvation resilience. Once appropriate antibodies become available, it will be interesting to use ChIP-Seq or ATAC-Seq approaches to narrow down the local action of Chm on specific genomic loci.

### Evolutionary implications and natural selection of *chm*

The temperature-sensitive phenotype of *chm* mutants raises intriguing questions about the evolutionary forces shaping its expression and function. Given that *Drosophila* in temperate regions experience seasonal temperature fluctuations, *chm* may have been selected to enhance survival during colder periods when metabolic rates are suppressed and starvation risk is heightened. This idea is supported by studies showing that *chm* exhibits a strong genetic variation in natural populations of *Drosophila* collected from either tropical or temperate habitats (Levine & Begun, 2008; Croze et al, 2017). These studies showed that flies from cooler habitats with a higher fluctuation in temperature have a much lower genetic variability than their tropical counterparts, suggesting a high evolutionary selection pressure on *chm* when environmental conditions change.

Paradoxically, although *chm* deficiency impairs starvation resistance at low temperatures, it has been reported that reduced *chm* activity can extend lifespan under nutrient-rich conditions (Peleg et al, 2016). This trade-off suggests that *chm* expression may be subject to balancing selection, where high expression promotes survival in fluctuating environments with periodic starvation and lower expression could be beneficial in stable, resource-abundant settings. Future studies examining *chm* allelic variation across geographically distinct *Drosophila* populations could provide further insight into how environmental pressures have shaped its evolutionary trajectory.

### Conclusion and future directions

Our findings position *chm* as a key epigenetic regulator that modulates starvation resilience in a temperature-dependent manner, likely by maintaining acetylation capacity under metabolically constrained conditions. The observation that *chm*'s role becomes critical only at lower temperatures underscores the intricate interplay between environmental cues, metabolism, and chromatin regulation. Future work should explore whether other histone-modifying enzymes exhibit similar temperature-sensitive phenotypes and whether metabolic intermediates (e.g., acetyl-CoA, NAD+) mediate these effects. In addition, investigating *chm*'s role in other stress responses, such as cold tolerance or oxidative stress, could reveal broader functions in environmental adaptation.

Finally, our study highlights the importance of considering environmental context when interpreting genetic phenotypes. The conditional requirement for *chm* underscores how epigenetic regulators can act as metabolic sensors, fine-tuning gene expression in response to external conditions. Understanding these mechanisms not only advances our knowledge of *Drosophila* physiology but may also provide insights into how metabolic–epigenetic interactions influence stress resilience in other organisms, including mammals.

## Materials and Methods

### Fly maintenance

Fly lines were maintained in an incubator (MLR-235H-PE; Panasonic) at 25°C with a 12/12-h light–dark cycle at 60% relative humidity. Composition of the fly food and comparison between Germany and Japan are given in Table S19. Details of fly lines used in the study are given in Table S20.

### Starvation assay

Flies of age 8–9 d developed, raised, and starved at corresponding temperatures with 60% relative humidity and 12-h light/12-h dark cycle were used for starvation experiments. Flies (30 flies—small vial; 50 flies—big vial) were transferred to an empty fly vial containing a tissue soaked in water (1 ml—small vial; 4 ml—big vial). The total number of flies for each experiment ranged from 100 to 200 for each genotype/condition tested. Readings were taken until flies were dead in all genotypes with dead flies counted every 10–12 h.

Experiments at 23°C and 25°C (in males) were performed in both Japan and Germany to exclude any differences in food and other laboratory conditions. For females, starvation at 23°C and 25°C was performed only in Germany. Furthermore, starvation at 18°C, 21°C, and 27°C in both males and females was performed only in Japan.

For starvation assay with supplementation, flies were developed, raised, and starved at 23°C. Around 3–4 d after eclosion, male and female flies were separated. For this experiment, only male flies were used. Male flies of age 8–9 d were then transferred to an empty bottle with a wet tissue containing only water (4 ml) or water and corresponding supplement (4 ml). Both sodium acetate and sodium citrate were used at a concentration of 125 mM.

### PTM of histones

#### *Sample preparation*
Fly heads were obtained following the same procedure given in the *Body and head weight measurement* section. Approx. 300 µl of

homogenization buffer (60 mM KCl, 15 mM NaCl, 4 mM MgCl$_2$, 15 mM Hepes [pH 7.5], 0.5% Triton X-100, 0.5 mM DTT, 20 mM sodium butyrate, and 1 tablet protease inhibitor) was added to 30–50 fly heads and was homogenized extensively with an electrical stirrer (5 × 10 s ON and 15 s OFF). After this, sonication was performed with Bioruptor Pico for 3 × 10 s ON and 45 s OFF at 4°C. The obtained lysate was centrifuged at 20,817$g$ for 30 min. The obtained pellet was resuspended in 200 $\mu$l of 0.2 M H$_2$SO$_4$, vortexed heavily, and rotated overnight at 4°C. Subsequently, the overnight incubated lysate was centrifuged at 20,817$g$ (max. Speed) for 10 min at 4°C. Histone was precipitated by adding trichloroacetic acid (TCA) (Cat. No. 85183; Thermo Fisher Scientific) to reach 26% final concentration. Tubes were mixed and incubated at 4°C for 2 h and spun at 20,817$g$ for 15 min. Pellets were washed thrice with ice-cold 100% acetone (Cat. No. AA22928-K2; VWR) (5-min rotation at 4°C, 15 min of 20,817$g$ spin at 4°C between washes), dried for 15 min at RT, and resuspended in 20 $\mu$l of 1x Laemmli sample buffer for million cells and boiled at 95°C for 5 min. Samples were stored at –20°C until further use. The histones corresponding to 0.5 million cells were separated onto 4–20% precast polyacrylamide gels (Cat. No. 43277.01; SERVA). Gels were briefly stained with InstantBlue Coomassie Protein Stain (Cat. No. ab119211; Abcam). For targeted mass spectrometry analysis, histones bands were excised, washed once with MS-grade water (Cat. No. 1153331000; Sigma-Aldrich), and de-stained twice (or until transparent) by incubating for 30 min at 37°C with 200 $\mu$l of 50% acetonitrile (ACN) (Cat. No. 8825.2; Carl Roth) in 50 mM ammonium bicarbonate (NH$_4$HCO$_3$) (Cat. No. T871.1; Carl Roth). Gel pieces were then washed twice with 200 $\mu$l MS-grade water and twice with 200 $\mu$l of 100% ACN to dehydrate them. Histones were in-gel–acylated by first adding 20 $\mu$l of d6 acetic anhydride (175641-5G; Sigma-Aldrich), followed by 40 $\mu$l of 100 mM NH$_4$HCO$_3$. After 5 min, 140 $\mu$l of 1 M NH$_4$HCO$_3$ was slowly added to the reaction. pH of the final solution should be around 7 (in cases where pH was acidic, few microliters of 1 M NH$_4$HCO$_3$ was added). Samples were incubated at 37°C for 45 min with contiuous shaking. After this, samples were washed five times with 200 $\mu$l of 100 mM NH$_4$HCO$_3$, four times with 200 $\mu$l of MS-grade water, and four times with 200 $\mu$l of 100% ACN. They were spun down briefly, and all remaining ACN was removed. Gel pieces were rehydrated in 50 $\mu$l of trypsin solution (25 ng/ml trypsin in 100 mM NH$_4$HCO$_3$) (Cat. No. V5111; Promega) and incubated at 4°C for 30 min. After the addition of 150 $\mu$l of 50 mM NH$_4$HCO$_3$, histones were in-gel–digested overnight at 37°C with continuous shaking. Peptides were sequentially extracted by incubating for 10 min at RT with 150 $\mu$l of 50 mM NH$_4$HCO$_3$, and washed twice with 150 $\mu$l of 50% ACN (in MS-grade water)/0.1% trifluoroacetic acid (TFA) and twice with 100 $\mu$l of 100% ACN. During each of the above washing step, samples were sonicated for 3 min in a water bath followed by a brief spin down. Obtained peptides were dried using a centrifugal evaporator and stored at –20°C until resuspension in 30 $\mu$l of 0.1% TFA. For desalting, peptides were loaded in a C18 StageTip (prewashed with 20 $\mu$l of methanol followed by 20 $\mu$l 80% ACN/0.1% TFA and equilibrated with 20 $\mu$l of 0.1% TFA) and washed two times with 20 $\mu$l of 0.1% TFA, and peptides were eluted three times with 10 $\mu$l of 80% ACN/0.25% TFA. Flow-through obtained from loading of peptides in C18 was further desalted with TopTip Carbon (Cat. No. TT1CAR.96; GlyGen) by loading the flow-through thrice (prewashed

thrice with 30 $\mu$l of 100% ACN followed by equilibration thrice with 30 $\mu$l of 0.1% TFA), washed five times with 30 $\mu$l of 0.1% TFA, and eluted thrice with 15 $\mu$l of 70% ACN and 0.1% TFA. Eluted peptides from both desalting steps were combined and evaporated in a centrifugal evaporator, resuspended in 15–17 $\mu$l of 0.1% TFA, and stored at –20°C until mass spectrometry analysis.

### *Targeted mass spectrometry*

Desalted histone peptides in 0.1% TFA were injected in an RSLCnano system (Thermo Fisher Scientific) and separated in a 15-cm analytical column (75 $\mu$m ID home-packed with ReproSil-Pur C18-AQ 2.4 $\mu$m from Dr. Maisch) with a 50-min gradient from 4% to 40% ACN in 0.1% formic acid at 300 nl/min flow rate. The effluent from the HPLC was electrosprayed into a Q Exactive HF mass spectrometer (Thermo Fisher Scientific). The MS instrument was programmed to target several ions. Survey full-scan MS spectra (from m/z 270–730) were acquired with resolution R = 60,000 at m/z 400 (AGC target of 3 × 10$^6$). Targeted ions were isolated with an isolation window of 0.7 m/z to a target value of 2 × 10$^5$ and fragmented at 27% normalized collision energy. Typical mass spectrometric conditions were as follows: spray voltage, 1.5 kV; no sheath and auxiliary gas flow; heated capillary temperature, 250°C (Pino et al, 2020).

### *Data analysis*

Raw data from mass spectrometry were analyzed using Skyline v21.1. Peak integration was performed for H3 and H4 peptides for each of its corresponding modifications. Relative levels of each PTM were calculated from the obtained intensities using R environment, based on the formula given in Feller et al (2015). Ellipses for PCA were drawn unsupervised using assuming multivariate t-distribution and bootstrapped statistics (check fviz_pca_ind() or ggplot2::stat_ellipse() function). Statistical tests for histone PTM analysis were based on a linear regression model. Corresponding significant *P*-values have been mentioned in the figures. As samples corresponding to both temperature conditions were processed simultaneously, they were directly compared across temperatures.

## RNA sequencing

### *RNA extraction*

30 heads were homogenized with an electrical stirrer with 500 $\mu$l of TRIzol (Cat. No. 15596026; Thermo Fisher Scientific). Chloroform was added at the ratio of 1:5 with TRIzol, and the solutions were mixed for 15 s by inverting the tubes. After 5-min incubation at RT, samples were centrifuged at 12,000$g$ for 15 min. The aqueous phase was transferred to a new tube from the centrifuged sample to which isopropanol was added at 1:1 ratio of the obtained aqueous phase, vortexed briefly, and incubated for 10 min at RT. They were then centrifuged for 10 min at 12,000$g$. The supernatant was discarded, and the obtained pellet was washed with 750 $\mu$l of 80% ethanol. After brief vortexing, samples were centrifuged at 8,000$g$ for 5 min. The obtained supernatant was discarded, and the pellet was air-dried for 5 min inside the hood and resuspended in RNase-free water.

### Library preparation

1 µg of RNA, obtained from fly heads, was used for library preparation. Both total and mRNA quality was assessed on 2100 Bioanalyzer (Cat. No. G2939BA; Agilent Technologies) using a RNA Pico Assay kit (Agilent RNA 6000 Pico Kit, Cat. No. 5067-1513) using the manufacturer's protocol. rRNA depletion was performed using NEBNext rRNA Depletion Kit (Human/Mouse/Rat) (#E6310; NEB), and library preparation for RNA sequencing was performed using NEBNext Ultra II Directional RNA Library Prep Kit for Illumina (#E7760; NEB) following the manufacturer's protocol. Libraries were sequenced on an Illumina HiSeq 1500 instrument at the Laboratory of Functional Genomic Analysis (LAFUGA, Gene Center Munich, LMU).

### Data analysis

A total of 50-bp paired-end reads were aligned to the *D. melanogaster* reference genome (release 6) using the STAR aligner (version 2.5.3a) with providing GTF annotation (dmel-all-r6.17.gtf). Reads with multiple alignments were filtered by setting outFilterMultimapNmax parameter to 1. Reads were counted per gene with parameter –quantMode GeneCounts. BAM files were converted to normalized bedgraph coverage using genomeCoverageBed command (bedtools version 2.27.1) with -scale parameter set to divide by the total number of reads and multiply by a million. Bedgraph files were converted to tdf files (igvtools version 2.3.98) to visualize in the IGV browser.

Count tables (read counts per gene) were read into R environment, and low count genes were filtered out (at least 3 reads per gene in 10% of the samples analyzed together). Differential expression analysis was performed by DESeq2 (Love et al, 2014) package (version 1.24) by adding replicate information as a batch variable. Samples that were directly compared with each other were fitted in the same DESeq2 model. Log$_2$FoldChange estimates and adjusted *P*-values were obtained by the results function (DESeq2), and an FDR cutoff and Π-value ≤ 0.05 were applied. In addition, the less stringent Π-value includes both statistical and biological information by considering log$_2$FoldChange and *P*-value to obtain values between 0 and 1 (Xiao et al, 2014; Hostrup et al, 2022). For PCA analysis, batch effect was corrected by the remove batch effects function from limma (Ritchie et al, 2014) (package version 3.52.0) on the normalized read counts.

Gene set enrichment analysis was performed on the obtained results from different conditions/comparisons using the gseGO function from clusterProfiler (Yu et al, 2012) (package version 3.12.0) by ranking the genes based on t-statistic value without any log$_2$-FoldChange or p-adjusted cutoff. GSEA plots with selected GO terms were also generated with R environment with these selected GO terms having an FDR ≤ 0.05 cutoff.

To compare the transcriptomic data from different temperatures, we used our already published data of control and *chm* mutant at fed and starved conditions at 23°C. All samples were normalized relative to their corresponding temperature control before comparison wherever necessary.

### Quantitative reverse transcriptase PCR

### RNA extraction

30 heads were homogenized with an electrical stirrer with 500 µl of TRIzol (Cat. No. 15596026; Thermo Fisher Scientific). Chloroform was added at the ratio of 1:5 with TRIzol, and the solutions were mixed for 15 s by inverting the tubes. After 5-min incubation at RT, samples were centrifuged at 12,000*g* for 15 min. The aqueous phase was transferred to a new tube from the centrifuged sample to which isopropanol was added at 1:1 ratio of the obtained aqueous phase, vortexed briefly, and incubated for 10 min at RT. They were then centrifuged for 10 min at 12,000*g*. The supernatant was discarded, and the obtained pellet was washed with 750 µl of 80% ethanol. After brief vortexing, samples were centrifuged at 8,000*g* for 5 min. The obtained supernatant was discarded, and the pellet was air-dried for 5 min inside the hood and resuspended in RNase-free water. RNA concentration and A260/280 ratio were measured with NanoDrop.

### DNase treatment, cDNA synthesis, and qRT–PCR

DNase treatment (Roche, DNase I recombinant, RNase-free from bovine pancreas, Cat. No. 04716728001) was performed with 1 µg of RNA as starting material following the manufacturer's instructions. cDNA synthesis was performed using SuperScript III First-Strand Synthesis System (Cat. No. 18080051; Invitrogen, Random hexamer priming) using DNase-treated RNA. Each reaction was set up with/ without SuperScript III reverse transcriptase. The obtained cDNA was treated with RNase H to remove the RNA-DNA duplex and diluted 1:5 with RNase-free water. Diluted cDNA was used for qRT-PCR with Fast SYBR Green Master Mix (Cat. No. 4385612; Thermo Fisher Scientific) following the manufacturer's instructions and run on a LightCycler 480 II (Roche) instrument. Primer efficiency was calculated using serial dilutions, and the corresponding melt curves were also assessed. Sequences of qRT-PCR primers used are given in Table S21.

### Weight measurement

Flies of the fed and starved conditions were transferred to a 2.0-ml tube and snap-frozen in liquid nitrogen. After this, the weight of each empty tube was measured. Heads and bodies were obtained by passing through sieves. First sieve (width: 710 µm) separates the bodies from remaining, and the second (width: 355 µm) separates heads from wings/legs (Analysensieb). 20–30 heads and bodies were transferred to the corresponding 1.5-ml tube, and the weight was measured again. The difference in weight between the two was considered as the corresponding weight of 20–30 flies from which weight per fly was calculated. Weights were measured using a KERN ABJ 120-4NM weighing machine. Samples and all the components were kept in dry ice for the entire duration of the experiment.

### Carbohydrate (glucose, glycogen, and trehalose) and TAG quantification

This methodology was adopted and modified from Tennessen et al (2014), with minor modifications. Briefly, 10–15 flies were snap-frozen in liquid nitrogen and lysed with 1x PBS (for carbohydrate) and 1x PBST 0.05% (for TAGs) using an electric stirrer. An aliquot from the obtained lysate was used for protein quantification using Quick Start

Bradford 1x Dye Reagent (Cat. No. 5000205; Bio-Rad). After heat treatment, the lysate was stored at −80°C until analysis.

For glycogen and trehalose, samples were treated either with amyloglucosidase (Cat. No. 11202332001; Sigma-Aldrich) or with porcine trehalase (Cat. No. T8778-1UN; Sigma-Aldrich). In addition, for both glycogen and trehalose, non–enzyme-treated control, to measure the basal glucose, was also used by replacing the corresponding enzyme with 1x PBS or 1x trehalase buffer (5 mM Tris, pH 6.6, 137 mM NaCl, 2.7 mM KCl). Accordingly, glucose, glycogen, and trehalose standards were prepared from 0 to 0.16 mg/ml range. For glycogen, samples were diluted between 1:3 and 1:5 and the enzymatic treatment was carried out for 60 min at 37°C, whereas for trehalose, samples were either undiluted or diluted 1:2 and the enzymatic treatment was carried out overnight at 37°C. After this, absorbance at 340 nm was measured using a plate reader. The amount of glycogen and trehalose was calculated by measuring the difference between enzyme-treated and non-treated samples and normalized to protein amounts. In cases where the absorbance falls below the standard curve, the value was replaced with zero as they were below the threshold limit of the quantification method.

For TAGs, samples were treated with triglyceride reagent (Cat. No. T2449; Sigma-Aldrich). In addition, non–enzyme-treated control, to measure the basal glycerol, was also used by replacing the corresponding enzyme with 1x PBST. Accordingly, glycerol standards were prepared from 0 to 1.0 mg/ml range using 2.5 mg/ml triolein equivalent glycerol standard solution (Cat. No. G7793; Sigma-Aldrich). For quantification of TAGs, samples were undiluted and the enzymatic treatment was carried out for 60 min at 37°C. After this, absorbance at 540 nm was measured using a plate reader. The amount of TAG was calculated by measuring the difference between enzyme-treated and nontreated samples and normalized to protein amounts. In cases where the absorbance fall below the standard curve, the value was replaced with zero as they were below the threshold limit of the quantification method.

## Metabolomics

### Sample preparation
Fly heads were obtained following the same procedure given in the *Body and head weight measurement* section.

### Mass spectrometry
Metabolomics profiling was conducted using ion-pairing reversed-phase liquid chromatography–high-resolution mass spectrometry (LC-HRMS) modified from previous methods for polar analytes (Guo et al, 2016; Kuskovsky et al, 2019). Samples were spiked with a stable isotope mix containing 13C3-sodium pyruvate, 13C3-lactate, 13C4-fumaric acid, 13C4-succinic acid, 13C415N1-aspartic acid, 13C6-citric acid, 13C6-glucose-6-phosphate, 13C5-D-$\alpha$-hydroxyglutaric acid, and 13C515N2-glutamine from Cambridge Isotope Laboratories. 1 ml of Optima LC-MS-grade 80:20 methanol:water prechilled to −80°C was then added to each sample, followed by a 30-s vortex mixing and 15-s pulse sonication with a probe tip sonicator, and then, samples were returned to the −80°C freezer for 30 min. Insoluble debris was precipitated by centrifugation for

10 min, at 17,000$g$ at 4°C. The supernatant was evaporated to dryness under nitrogen and resuspended in 100 $\mu$l 5% 5-sulfosalicylic acid in water, and 5 $\mu$l was injected for analysis. LC-HRMS was conducted on an Ultimate 3000 UHPLC equipped with a refrigerated autosampler (at 6°C) and a column heater (at 55°C) with a HSS C18 column (2.1 × 100 mm i.d., 3.5 $\mu$m; Waters) used for separations. Solvent A was 5 mM DIPEA and 200 mM HFIP, and solvent B was methanol with 5 mM DIPEA, 200 mM HFIP. The gradient was as follows: 100% A for 3 min at 0.18 ml/min, 100% A at 6 min with 0.2 ml/min, 98% A at 8 min with 0.2 ml/min, 86% A at 12 min with 0.2 ml/min, 40% A at 16 min, and 1% A at 17.9–18.5 min with 0.3 ml/min, then increased to 0.4 ml/min until 20 min. Flow was ramped down to 0.18 ml/min back to 100% A over a 5-min re-equilibration. For MS analysis, the UHPLC was coupled to a Q Exactive HF mass spectrometer (Thermo Fisher Scientific) equipped with a HESI II source operating in negative mode. The operating conditions were as follows: spray voltage 4,000 V; vaporizer temperature 200°C; capillary temperature 350°C; S-lens 60; in-source CID 1.0 eV, resolution 60,000. The sheath gas (nitrogen) and auxiliary gas (nitrogen) pressures were 45 and 10 (arbitrary units), respectively. Single ion monitoring (SIM) windows were acquired around the [M-H]- of each analyte with a 20 m/z isolation window, 4 m/z isolation window offset, 1 × $10^6$ AGC target, and 80 ms IT, alternating in a full MS scan from 70 to 950 m/z with 1 × $10^6$ AGC and 100 ms IT. Data were analyzed in Xcalibur v4.0 and/or TraceFinder v4.1 (Thermo Fisher Scientific) using a 5-ppm window for integration of the peak area of all analytes and internal standards used for normalization.

### Data analysis
As samples corresponding to both temperature conditions were processed simultaneously, they were directly compared across temperatures. R Core Team (2021) was used for the analysis of metabolomics data. We also included filtering criteria to remove metabolites with a lot of missing values. For further comparison and analysis, we considered metabolites that had less than the three missing values across conditions and replicates.

## Food consumption

### Preparation of agar-based fly food with 0.5% brilliant blue FCF
This methodology was adopted and modified from Shell et al (2018). The recipe shown in Table S22 is for 1 liter of the agar-based fly food. Agar and distilled water were mixed by heating and stirring (use magnetic stirrer and heater) them continuously until agar is dissolved.

Once agar is dissolved, apple juice and Rubensirup were added and mixed well. After this, brilliant blue FCF was also added, and all these components were mixed until dissolved. Once the temperature reaches around 70°C, Nipagin was added little by little with constant stirring. Approximately 9 ml of the fly food was added to a 60-mm petri dish and kept at RT (around 25°C) until solidified. The petri dishes were then stored at 4°C until use. Before transferring the flies for experiment, the food was kept at RT for at least 1 h.

Measurement of consumed and excreted food: Once the food is warm, 15–20 flies were transferred to an empty fly vial, and the open end was sealed with the petri dish containing fly food. Cellotape was used to make sure that the petri dish does not get disturbed. The flies were then kept at 23°C, with a 12/12-h light–dark cycles and 60% humidity for 24 h. Flies were then collected and transferred to an Eppendorf tube. 200 $\mu$l of $dH_2O$ was added and homogenized using electrical homogenizer, followed by centrifugation at full speed for 5 min. The supernatant was transferred, and the insoluble material was discarded. The obtained supernatant was made up the volume to 1.5 ml with $dH_2O$. To measure the amount of food excreted, 5 ml of $dH_2O$ was added to the sides of the fly vial and pipetted up and down with a 1-ml pipette to make sure all the colored excreta were resuspended, and the obtained solution was transferred to an Eppendorf tube. Both intake and excreted food absorbance was measured at 630 nm in a plate reader with $dH_2O$ as blank. To measure total food consumed, the absorbance from intake and excreted food was added.

### Plots and statistical analysis

All statistical analysis was performed in R Core Team (2021) environment unless otherwise mentioned. Statistical tests were decided based on the experimental design and measurements and have been mentioned in each of the figure legends.

Statistical analysis for weight (or weight loss) quantification, metabolite measurements (both colorimetric and metabolomics), and supplementation was performed using ANOVA by considering all possible interaction terms, followed by $P$-value adjustment based on interaction terms, wherever necessary. This strategy was employed to assess the interaction between different confounding factors such as temperature, genotype, and/or starvation time on the corresponding dependent variable. This type of statistical analysis term will provide information not just about individual contributions of each factor but also the effect of these factors on each other. Depending on the number of confounding factors, we performed either a three-way or two-way ANOVA using the following formula:

(i) Dependent variable ~ Genotype * Supplementation + Error (Batch) (for acetate/citrate supplementation, paired data) and Dependent variable ~ Genotype * Time + Error (Batch) (for *chm* overexpression data).

(ii) Dependent variable ~ Genotype * Temperature * Time_group + Error (Batch) (colorimetric assays, paired data).

(iii) Dependent variable ~ Genotype * Temperature * time_point (metabolomics, unpaired data).

Graphics in all figures were created using Biorender.com, whereas plots and graphs for visualization were generated in R environment unless otherwise mentioned.

## Data Availability

The datasets produced in this study are available in the following databases: *Transcriptomic data (23°C)*: Venkatasubramani et al (2023), Gene Expression Omnibus GSE211042; *Transcriptomic data (25°C)*: Gene Expression Omnibus GSE287187; *Histone PTM mass spectrometry data (23°C and 25°C)*: Venkatasubramani et al (2023), ProteomeXchange Consortium via the PRIDE[20] partner repository PXD035947.

## Supplementary Information

## Acknowledgements

We would like to thank the members of the Imhof laboratory, Peleg laboratory, Tanimoto laboratory, Becker department, Andreas Ladurner, and Raffaele Teperino for their inputs and suggestions, Catherine Regnard and Silke Krause from Becker department, Stefan Krebs and Helmut Blum from LAFUGA facility for sequencing, and Nicolas Gompel and Christa Schwarzlose for fly food and maintenance, respectively. We thank Tobias Straub, Tamas Schauer, and Wasim Aftab for their assistance in experimental design, statistics, and bioinformatics analysis. We thank Markus Hohle from QBM for his constant support. AV Venkatasubramani was supported by the QBM and SFB1309. T Ichinose was supported by the Ministry of Education, Culture, Sports, Science and Technology (MEXT): 21K06369. Work in the AI laboratory was funded by grants from the DFG, grant numbers 213249687 (CRC1064) and 325871075 (CRC1309). The Peleg laboratory was supported by the FBN, DFG grant (458246576), and Longevity Impetus grant from Norn Group.

### Author Contributions

AV Venkatasubramani: conceptualization, data curation, formal analysis, investigation, visualization, and writing—original draft.
T Ichinose: investigation and methodology.
I Forne: data curation, formal analysis, investigation, and methodology.
NW Snyder: funding acquisition, investigation, and methodology.
H Tanimoto: supervision, funding acquisition, methodology, and project administration.
S Peleg: conceptualization, supervision, and writing—original draft, review, and editing.
A Imhof: conceptualization, resources, supervision, funding acquisition, visualization, project administration, and writing—original draft, review, and editing.

### Conflict of Interest Statement

S Peleg is a cofounder of Luminova Biotech. Other authors have nothing to declare.

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
