## [Reviewer comments · Life Science Alliance]

Chameau (HBO1) regulates starvation resistance in *Drosophila melanogaster* in a temperature-dependent manner

Anuroop Venkateswaran Venkatasubramani, Toshiharu Ichinose, Ignasi Forne, Nathaniel Snyder, Hiromu Tanimoto, Shahaf Peleg, and Axel Imhof

DOI: <https://doi.org/10.26508/lsa.202503524>

Corresponding author(s): Axel Imhof, Ludwig-Maximilians-Universität München and Shahaf Peleg, Research Institute for Farm Animal Biology

Review Timeline:

Submission Date:	2025-10-08
Editorial Decision:	2025-10-13
Revision Received:	2025-10-21
Accepted:	2025-10-23

Scientific Editor: Tim Fessenden

Transaction Report:

Please note that the manuscript was reviewed at *Review Commons* and these reports were taken into account in the decision-making process at *Life Science Alliance*.

Review
COMMONS

Reviews

Review #1

Evidence, reproducibility and clarity

****Summary:****

In this study, the authors of the manuscript investigated the role of an acetyltransferase - Chm in starvation response at different temperatures using fruit fly mutant lines. The authors started with testing survival in wildtype and mutant Chm fruit flies upon starvation at different temperatures. They found that fruit flies have lower starvation resilience with lower Chm expression at lower temperatures, while at higher temperatures, the lower Chm expression does not affect starvation resilience. Then they further analysed the histone acetylation and methylation, and transcriptome in wildtype and mutant Chm fruit flies at different temperatures, which further led them to investigate the metabolism of the flies including body weight, glucose levels, and trehalose levels. They concluded that Chm has a role in regulating the starvation resilience at different temperatures. This is quite interesting, especially considering there appears to be habitat-temperature dependent variation in Chm in the wild, reported by others.

****Major comments:****

- The authors claim that an increase of 2{degree sign}C was sufficient to restore starvation resilience in Chm mutants. However, in the survival experiment, the survival of both control and Chm mutant flies at 25{degree sign}C were lower than at 23{degree sign}C, so we would think that it is more likely that the effect of chm is collapsed or overwhelmed by the temperature change rather than restored. Is Chm rather involved in temperature tolerance or temperature-dependent metabolic regulation? A more nuanced discussion and conclusion are needed.
- There are parts where data from a previous paper and new data are combined, for example, in Figure 1, the starvation survival of the Chm mutant lines at 23{degree sign}C was previously published. The authors need to make it explicit when they are including data that was already published. If the experiments were not done at the same time, they cannot be compared.
- When using Chm RNAi line and Chm mutant line, at least some validation needs to be done to show the expression level of Chm, with e.g. qPCR.
- For the PCA analysis of the histone acetylation (line 126-127) in Figure 2, was the separation manually performed or was there any clustering methods performed with PCA results like KNN? Please describe how the broad areas (circles) were decided and if simply by eye, the authors should remove them. Please also make it clear if they are fed or starved flies or both.
- This whole analysis of histone acetylation and methylation does not include any statistics - the authors should show statistically significant differences for at least some of the relevant modifications. The number of biological replicates here is also not clear. Please give sufficient information in the results section about how this experiment was conducted.
- In line 155, regarding the significant genes, please make it clear what comparisons they made giving those significant genes.
- Regarding RNA-Seq: what was done is not clear from the results. They should make it clear. Why did the authors analyse head RNA? How can this tell them about metabolism of the whole fly - is head an important metabolic tissue/part of the fly? Please note FDR in results section.
- We think the link between Chm and histone acetylation, between Chm and the starvation response could be strengthened with the use of Chm overexpression line.
- It's quite unclear to us how the transcriptome changes lead to different starvation resilience at different temperatures with or without Chm. Please identify and validate a few key genes in the transcriptome change and their influence on the corresponding behaviours in the context of starvation response.
- We do wonder if there are big differences between the fed groups across different conditions regarding the RNA-Seq data.

****Minor comments:****

- In line 87, when using acronym PTMs for the first time please also write in full.
- In line 135, what does "these processes" mean?
- In line 176, "Chm has a much stronger effect on gene expression at 23{degree sign}C than at 25{degree sign}C" should be clearer with the context of starvation resilience. Otherwise please show direct differential analysis between the flies at two temperatures.

- In 182, please explain more about the drug and how it is relevant to this study.
- In Figure 4, it's interesting to see the control flies at 23°C could build up the trehalose. Would this be related to why control flies at 23°C were the longest-lived group?
- The numbers of Supplemental Figures 3-6 are wrong, please fix.

2. Significance:

The authors did a good job performing a series of logical experiments to show Chm is relevant to histone acetylation and is potentially associated with metabolic regulation that has different effect/occurs at different temperatures, which provide insight into basic mechanisms of how metabolism and the starvation response at different temperatures are regulated. However, we think the details of some methods and statistics need to be improved, as we point out in the specific comments above. The relationships between key elements in this study, including Chm, histone acetylation, and metabolism pathways, could be made more solid and the mechanisms clarified. There is an interesting angle, regarding significance, that is somewhat underdeveloped: the variation in Chm allele frequency in the wild.

Review #2

1. Evidence, reproducibility and clarity:

The manuscript submitted by Venkatasubramani et al., is a follow up of their previous paper published in EMBO Reports, which showed quite elegantly the role of Chm in regulating lifespan, its effect on starvation responses in *Drosophila*. The previous paper also showed gene expression changes/protein modifications that are affected in the absence of Chm, which were responsible for the phenotypes seen.

****Main findings of the manuscript:****

1. Lack of Chm leads to starvation sensitivity at 23°C and below. However, at higher temperatures of 25°C this survival impairment was not visible. Shows that Chm has no role in resisting the effects of starvation at higher temperatures.
2. Food consumption was not affected in *chmMYST/+* mutants at both 23°C and 25°C. The basal energy/hunger levels are not regulated by Chm at lower and higher temperatures.
3. H4 acetylation showed distinct changes based on the temperature shift of 2°C and lack of Chm. H4K12-ac levels were lower in *chmMYST/+* mutants, but H4 di-ac was affected by both genotype and temperature. Thus, while some roles of Chm on H4 acetylation are temperature specific, some others are not.
4. Chm regulates the reduction of H4K12-ac/H4 di-ac levels in response to starvation at 23°C, whereas, the reduction of these acetylation events in response to starvation seems to be independent of Chm at 25°C. Hence, other factors are responsible for reduction of H4K12/H4 di acetylation in response to starvation at higher temperatures. H3-acetylation was affected only by temperature, and did not change during starvation or due to the lack of Chm. H3-methylation levels did not get affected by both temperature and lack of Chm, and the response to starvation has not been checked.
6. Analysis of transcription response to starvation showed distinct clustering based on genotype and nutrient status at 25°C, similar to 23°C. Similar transcriptional changes were observed in control and mutant flies, in response to starvation.
7. Genes differentially regulated in controls were encoding for proteins involved in carbohydrate metabolism at 25°C, and mitochondrial processes at 23°C. In mutants insulin, hormone and starvation response genes were affected more at 25°C, and metabolic processes at 23°C. Also, lack of *chm* caused a stronger effect on gene expression at 23°C in comparison to 25°C.
8. Chm is required for maintaining proper body weight at 23°C, and not at 25°C. Weight loss during starvation requires Chm at 23°C. However, maintaining body weight or weight loss in response to starvation does not require Chm at 25°C.
9. Differences in glucose levels between control and *chmMYST/+* flies were not affected by temperature, nor the utilization of glucose in response to starvation. Other nutrient stores TAG and glycogen were not affected by the lack of Chm or the temperature. While Chm is needed for maintaining normal Glucose levels at both lower and higher temperatures, TGA and Glycogen levels are not affected.
10. Chm maintains trehalose levels under control and trehalose levels did not increase in response to starvation when compared to controls, at 23°C. However, at 25°C, the levels of trehalose were not different between the genotypes and their levels did not increase in response to starvation even in control flies.
11. The levels of various trehalose metabolism proteins were found to be affected by the lack of Chm, also the acetylation status of these proteins were reduced at 23°C.

12. The fly weight differences and the reduced weight loss in response to starvation can be rescued by overexpression of Chm in the *chm* mutant background at 23°C. High trehalose levels seen at 23°C in *chm* mutants is also rescued, so is the change in trehalose levels in response to starvation. This confirms the role of Chm in regulation of body weight in response to starvation and trehalose levels at 23°C.

****Major comments:****

1. Many of the conclusions are vague and in many places do not match the evidence. The authors need to conclude with more clarity each sub section of their results. Also, state only what is being substantiated by experimental evidence.
2. No suggestions for any additional experiments, however serious rewriting is needed to make the manuscript more comprehensible.
3. I couldn't see any comments on the levels of Chm mRNA/protein in response to starvation or temperature. This would be an important point to address, they should show data atleast on the mRNA levels in these contexts.

****Minor comments:****

Supplementary figure numbers should be corrected.

2. Significance:

The current manuscript builds upon the previous paper but but does not provide a significant advance in the understanding of the role of Chm, except for some incremental information. The manuscript does not present a convincing story due to the lack of experimental proof, unclear presentation of the results and lack of coherence in its structure. General assessment:

1. Suppression of starvation sensitivity phenotype was observed even when *chm*MYST/+ flies were partially exposed to 25°C at developmental, adult or starvation stages. What does this mean in terms of the function of Chm protein and its effect on starvation stress response?
2. Apart from documenting evidence towards the effects of Chm and temperature on H4 acetylation status it is hard to conclude their significance on the physiology or metabolic status. Does this difference account to the survival of *chm*MYST/+ flies to starvation at 25°C? The levels of H4 acetylation in the *chm* mutants is very low at both 23°C and 25°C, in both early and late stages of starvation. Hence, while the claim that temperature shift from 23°C to 25°C can affect partially abrogate the effect of starvation on the reduction of H4K12-ac and H4 di-ac, it is not clear if this contributes to the survival of *chm* mutants during starvation at 25°C. While this section documents the role of Chm and temperature in H4 acetylation, and the influence of these factors on response of H4 acetylation during starvation, it fails to provide any significant improvement in understanding the functions of Chm in terms of the already published phenotypes.

The manuscript suggests that histone acetylation, rather than methylation, serves as a metabolic sensor in flies. However, the rationale behind this claim is unclear.

3. While the changes in transcription profile are interesting it remains unclear how this relates to the temperature dependence of starvation responses and other metabolic effects. This should be explained and discussed.
4. Weight loss, which could indicate nutrient expenditure during starvation, requires Chm only at 23°C. However, maintaining body weight or weight loss in response to starvation does not require Chm at 25°C. This indicates that Chm function is crucial for nutrient expenditure at 23°C, and not at 25°C, which could also explain their sensitivity to starvation at 23°C, and not at 25°C. This point is not discussed properly in the manuscript. Also, does this connect to the H4 acetylation status and gene expression changes in Chm mutants at lower and higher temperatures. The authors should discuss this.
5. While the effects on trehalose levels is affected by lack of Chm only at 23°C, this may not explain if this contributes to their starvation sensitivity at lower temperatures. Also, the induction of trehalose levels in response to starvation seems to be limited only to at 23°C. So the overall relevance of trehalose level change during starvation is not clear here.
6. The authors don't give any information regarding the changes in the levels of various trehalose metabolism proteins being responsible for the higher trehalose levels in *chm* mutants, or speculate if it contributes to the starvation sensitivity of *chm* mutants at 23°C.

Advances:

The manuscript clearly shows changes to starvation sensitivity, acetylation status, body weight changes, metabolic responses under the influence of Chm, and modified by the temperature during starvation responses. Some of which was reported earlier as well. Hence, the manuscript achieves its purpose if documenting these changes in these specific contexts was the original aim, without providing any mechanistic improvement in our understanding. Rewriting the manuscript to fulfil this purpose would help immensely.

Audience:

Broad, basic research, will be used for for readers who are interested in metabolic homeostasis

My field of interest:

Drosophila, insulin signalling, nutrient homeostasis, starvation responses, gene expression

Review #3

1. Evidence, reproducibility and clarity:

Summary:

In the manuscript titled "Chameau (HBO1) Regulates Starvation in a Temperature-Dependent Manner in *Drosophila melanogaster*," Venkatasubramani et al. investigated the role of the acetyltransferase Chm in regulating starvation in fruit flies under varying temperatures. They discovered that Chm becomes less effective at higher temperatures. This observation is particularly intriguing in the context of rising global temperatures due to climate change. However, a major shortcoming of the study is the lack of mechanistic insights behind this observation, which leaves the manuscript wanting.

Major comments:

1. The authors found that H4K12ac levels were significantly lower in *chm*MYST/+ flies and that H4 di-acetylation was influenced by temperature. It would be beneficial to conduct ChIP-seq analysis of these marks to see if they correlate with the differentially expressed genes in the authors' RNA-Seq datasets.
2. While the findings are interesting, the data do not provide insights into why an acetyltransferase might affect the observed phenotype. The authors attempt to discuss this under the section titled "What Can Cause the Temperature-Specific Changes in Histone Acetylation Levels?" However, it would be helpful to include supporting experimental evidence.
3. An illustration depicting a model of the underlying processes would be useful.

Minor comments:

4. Overall, the manuscript is well-written and easy to follow. However, there are some confusing statements. For instance, the authors write, "which was even larger at lower temperatures" (lines 94-95). It should be clarified what exactly is larger.
5. More detailed information should be included in certain areas to prevent readers from needing to reference previously published work. For example, the authors state, "Although male and female fruit flies have differences in starvation susceptibility, they show a similar temperature dependency." What specific differences in starvation susceptibility do they have?
6. Instead of using terms like "lighter," it would be more appropriate to use "lower weight" for scientific writing.

2. Significance:

The study presents compelling findings, but it is hindered by a lack of mechanistic insights that would deepen our understanding of the underlying processes. While the research has been conducted exclusively on fruit flies, a well-established model organism, the observations could resonate with a wide range of researchers and enthusiasts alike, given their potential implications for broader biological contexts.

We would like to thank the reviewers for their careful reading of our manuscript and their appreciation of our work. After a careful assessment of their comments, we realized that most of them were caused by misleading or insufficient explanations of the experiments or a somewhat short discussion. We therefore believe that we can address most if not all of the reviewers' comments by editorial changes. Below we provide a point-by-point reply on how we plan to address the points raised by the referees. Our comments to the reviewers' points are displayed in italics/blue.

Reviewer #1 (Major comments):

- ✓ The authors claim that an increase of 2°C was sufficient to restore starvation resilience in Chm mutants. However, in the survival experiment, the survival of both control and Chm mutant flies at 25°C were lower than at 23°C, so we would think that it is more likely that the effect of chm is collapsed or overwhelmed by the temperature change rather than restored. Is Chm rather involved in temperature tolerance or temperature-dependent metabolic regulation? A more nuanced discussion and conclusion are needed.

We thank the reviewer for this insightful comment. We agree that our original statement in the abstract was too categorical and did not sufficiently reflect the complexity of the data. We have therefore revised the sentence to read:

“Strikingly, the metabolic increase associated with a 2°C temperature rise was sufficient to bypass the requirement for Chm in starvation resilience, suggesting that Chm modulates metabolism in a temperature-dependent manner.”

This revised version reflects our interpretation that the observed resilience in chm mutants at 25°C is not due to a restoration of Chm function per se, but rather due to elevated metabolic activity at higher temperatures compensating for the reduction in Chm levels.

To further support this conclusion and address the reviewer's suggestion, we have added a new experiment demonstrating that supplementation with metabolites that increase intracellular acetyl-CoA levels (citrate or acetate (new Fig 6E and F)) is also sufficient to bypass the starvation sensitivity of chm mutants. These findings strengthen our conclusion that Chm is part of a nutrient- and metabolism-sensitive regulatory network, and that its role becomes dispensable when acetyl-CoA levels are elevated either through increased temperature or direct metabolic supplementation.

We have expanded the discussion to incorporate these findings and now more explicitly consider the possibility that Chm acts in a temperature-dependent metabolic context, where its requirement is conditional on baseline metabolic activity. These changes are reflected in the revised manuscript (see p10-12).

- ✓ For the PCA analysis of the histone acetylation (line 126-127) in Figure 2, was the separation manually performed or was there any clustering methods performed with PCA results like KNN? Please describe how the broad areas (circles) were decided and if simply by eye, the authors should remove them. Please also make it clear if they are fed or starved flies or both.

Ellipses for PCA were drawn unsupervised using assuming multivariate t-distribution and bootstrapped statistics. We apologize for not describing the clustering method sufficiently and will include this in the legend of figure 2 in the revised manuscript.

- ✓ We think the link between Chm and histone acetylation, between Chm and the starvation response could be strengthened with the use of Chm overexpression line.

In our 2023 manuscript we showed that ubiquitous as well as tissue-specific overexpression of chm improves survival upon starvation (Fig. 6A-F, (Venkatasubramani et al, 2023)). In this manuscript we now show that the overexpression of chm indeed full rescues all the phenotypes we observed at 23°C (Increased trehalose levels, weight loss upon starvation etc. (page 9 of the revised manuscript). Interestingly, we did not observe a full rescue of the bulk histone acetylation by Chm overexpression suggesting that the effect of an ectopic Chm expression is also mediated by additional substrates as we have shown previously.

Reviewer #1 (Significance (Required)):

- ✓ The authors did a good job performing a series of logical experiments to show Chm is relevant to histone acetylation and is potentially associated with metabolic regulation that has different effect/occurs at different temperatures, which provide insight into basic mechanisms of how metabolism and the starvation response at different temperatures are regulated. However, we think the details of some methods and statistics need to be improved, as we point out in the specific comments above. The relationships between key elements in this study, including Chm, histone acetylation, and metabolism pathways, could be made more solid and the mechanisms clarified. There is an interesting angle, regarding significance, that is somewhat underdeveloped: the variation in Chm allele frequency in the wild.

As mentioned above, we improved the method and statistics section and completely overhauled the manuscript to clarify our hypothesis, which is now also illustrated in a graphic model shown in panel G of figure 6. We agree with the reviewer that the chm allele frequency in the wild is very interesting, which is why we discussed it more extensively in the revised manuscript in a dedicated paragraph headed: "Evolutionary Implications and Natural Selection" of chm

Reviewer #2 (Major comments):

- ✓ Many of the conclusions are vague and in many places do not match the evidence. The authors need to conclude with more clarity each sub section of their results. Also, state only what is being substantiated by experimental evidence. No suggestions for any additional experiments, however serious rewriting is needed to make the manuscript more comprehensible.

We have now completely overhauled the manuscript to improve clarity and coherence

- ✓ I couldn't see any comments on the levels of Chm mRNA/protein in response to starvation or temperature. This would be an important point to address, they should show data atleast on the mRNA levels in these contexts.

Temperature or starvation has no effect on the levels of Chm mRNA. We have already published the effect starvation in the previous manuscript (Fig EV4A, (Venkatasubramani et

al, 2023)) and will add the figure comparing *chm* levels at different temperatures in the revised manuscript (see data below).

Reviewer #2 (Significance (Required)):

- ✓ Suppression of starvation sensitivity phenotype was observed even when *chmMYST/+* flies were partially exposed to 25°C at developmental, adult or starvation stages. What does this mean in terms of the function of Chm protein and its effect on starvation stress response?

We believe that higher temperatures result in higher metabolic rates. At 25°C, we observed a significant increase in citrate, a precursor of cytosolic and nuclear acetyl-CoA. This increase will likely compensate for the lower levels of Chm. Therefore, exposure to 25 °C overcomes the phenotype, as higher levels of acetyl-CoA at 25 °C ensure physiological levels of protein acetylation despite lower levels of Chm. We have included these data in the revised manuscript. Our hypothesis is further supported by a set of experiments in which we compensated for the Chm phenotype at 23 °C by adding citrate to the water used in the starvation experiment. The results of these experiments are shown in the new figure 6 of the manuscript.

- ✓ Apart from documenting evidence towards the effects of Chm and temperature on H4 acetylation status it is hard to conclude their significance on the physiology or metabolic status. Does this difference account to the survival of *chmMYST/+* flies to starvation at 25oC? The levels of H4 acetylation in the *chm* mutants is very low at both 23oC and 25oC, in both early and late stages of starvation. Hence, while the claim that temperature shift from 23oC to 25oC can affect partially abrogate the effect of starvation on the reduction of H4K12-ac and H4 di-ac, it is not clear if this contributes to the survival of *chm* mutants during starvation at 25oC. While this section documents the role of Chm and temperature in H4 acetylation, and the influence of these factors on response of H4 acetylation during starvation, it fails to provide any significant improvement in understanding the functions of Chm in terms of the already published phenotypes. The manuscript suggests that histone acetylation, rather than methylation, serves as a metabolic sensor in flies. However, the rationale behind this claim is unclear.

As described in our response to reviewer1, we think that the higher temperature results in a higher metabolic rate, which in turn affects the kinetics of chm mediated acetylation of various proteins as exemplified on the levels of H4 acetylation. We have better explained our hypothesis in the revised manuscript and included a model in the new figure 6. The claim that histone acetylation rather than histone methylation serves as a metabolic sensor is based on the fact that we did not detect changes in H3 methylation at different temperatures or in chm^{MYST/+} flies or during starvation

- ✓ While the changes in transcription profile are interesting it remains unclear how this relates to the temperature dependence of starvation responses and other metabolic effects. This should be explained and discussed.

Consistent with previous experiments (Klepsatel et al, 2019; Jang & Lee, 2018), flies respond very differently to starvation depending on the surrounding temperature, which is what we observe as well (Fig 3). However, at 23°C only wildtype flies mount a full transcriptional response whereas both mutant and wildtype flies respond similarly to starvation at 25°C. We have now better explained this finding in the second paragraph of the results section and in the first paragraph of the discussion.

- ✓ Weight loss, which could indicate nutrient expenditure during starvation, requires Chm only at 23oC. However, maintaining body weight or weight loss in response to starvation does not require Chm at 25o. This indicates that Chm function is crucial for nutrient expenditure at 23oC, and not at 25oC, which could also explain their sensitivity to starvation at 23oC, and not at 25oC. This point is not discussed properly in the manuscript. Also, does this connect to the H4 acetylation status and gene expression changes in Chm mutants at lower and higher temperatures. The authors should discuss this.

As stated above, we believe that the temperature differences are caused by an interplay between metabolic pathways that produce acetyl-CoA and Chm levels. Importantly, Chm is present at similar levels in both temperatures (homozygous chm is lethal). We hypothesise that CHM plays a key role in mediating starvation and protein acetylation at both temperatures. However, while metabolic pathways can provide sufficient metabolites, such as acetyl-CoA (the precursor for histone acetylation), at 25 °C, we hypothesise that lower metabolic rates at colder temperatures, combined with lower levels of acetyl-CoA, result in impaired survival after starvation. This can be seen, for instance, in histone H4 measurements, where the level of acetylation drops faster in chm mutants at 23°C than in chm mutants at 25°C during starvation. We now discuss this in the first paragraph of the discussion.

- ✓ While the effects on trehalose levels is affected by lack of Chm only at 23°C, this may not explain if this contributes to their starvation sensitivity at lower temperatures. Also, the induction of trehalose levels in response to starvation seems to be limited only to at 23°C. So the overall relevance of trehalose level change during starvation is not clear here.

At 23°C, Trehalose levels are substantially higher in chm mutants even under fed conditions. In contrast to wildtype flies where trehalose levels raise upon starvation, chm mutants are unable to increase their (already high) Trehalose levels. At 25°C, trehalose levels are similarly low for both control and chm mutant under fed conditions and respond in a similar way to

starvation, which is probably why both genotypes have similar survival rates. In the revised manuscript we now point this out in the results section and in the discussion (p8 and p11)

Reviewer #2 (Advances):

The manuscript clearly shows changes to starvation sensitivity, acetylation status, body weight changes, metabolic responses under the influence of Chm, and modified by the temperature during starvation responses. Some of which was reported earlier as well. Hence, the manuscript achieves its purpose if documenting these changes in these specific contexts was the original aim, without providing any mechanistic improvement in our understanding. Rewriting the manuscript to fulfil this purpose would help immensely.

As mentioned above. we completely overhauled the manuscript to address the point raised by all reviewers.

Reviewer #2 (Minor comments):

- ✓ More detailed information should be included in certain areas to prevent readers from needing to reference previously published work. For example, the authors state, "Although male and female fruit flies have differences in starvation susceptibility, they show a similar temperature dependency." What specific differences in starvation susceptibility do they have?

We observed that females exhibit similar phenotypic responses albeit at a different temperature range compared to males, where starvation at even higher temperatures eliminates the survival differences between genotypes. This may be attributed to sexual dimorphism in Drosophila. Females generally consume more food, have higher lipid (TAG) stores, and break down TAGs more slowly during starvation than males (Kubli, 2010; Lee & Jang, 2014; Wong et al, 2009). Additionally, dietary changes lead to sex-specific alterations in lipid content and gene expression (Groef et al, 2021; Wat et al, 2020). While this study primarily focused on males for mechanistic analysis, investigating sex-specific molecular responses would be a valuable direction for future research. However, as both genders show similar general behavior with regards to the role of chm, we removed this section from the manuscript to maintain clarity.

Reviewer #1 (Major comments):

- ✓ There are parts where data from a previous paper and new data are combined, for example, in Figure 1, the starvation survival of the Chm mutant lines at 23°C was previously published. The authors need to make it explicit when they are including data that was already published. If the experiments were not done at the same time, they cannot be compared.

This information is provided in the legends and in the main text, where necessary. We have cited our previously published manuscript to indicate all data that was obtained from previous study. We state this more clearly in the revised manuscript

- ✓ This whole analysis of histone acetylation and methylation does not include any statistics the authors should show statistically significant differences for at least some of the relevant modifications. The number of biological replicates here is also not clear. Please give sufficient information in the results section about how this experiment was conducted.

All p-values that are significant have been mentioned in the figure for histone analysis in fig.2. The legend and method section also specifies the type of statistical testing that has been used to calculate the p-values. We made sure to include all the number of biological replicates in each of the figure legends of the revised manuscript and expand the details of the statistical analysis in the method section.

- ✓ In line 155, regarding the significant genes, please make it clear what comparisons they made giving those significant genes.

We completely overhauled the section describing our results of the transcriptomic effects of chm in a new section entitled: "Transcriptomic effects of chm are temperature dependent"

- ✓ Regarding RNA-Seq: what was done is not clear from the results. They should make it clear. Why did the authors analyse head RNA? How can this tell them about metabolism of the whole fly - is head an important metabolic tissue/part of the fly? Please note FDR in results section.

The reasoning for using of head in transcriptomics will be added to the revised manuscript. FDR information is now provided in the legends and methods and in the results section of the revised manuscript of the manuscript

- ✓ We do wonder if there are big differences between the fed groups across different conditions regarding the RNA-Seq data.

The differences between genotypes under fed conditions are minimal irrespective of the temperature. This can be seen from PCA (Fig. S8B for 25°C and Fig. 5B in (Venkatasubramani et al, 2023) for 23°C) which shows only 10% variation in PC2, which corresponds to genotypic differences. There were approx. 100 genes which were significantly different although the fold-changes were very small.

Reviewer #1 (Minor comments):

- ✓ In line 87, when using acronym PTMs for the first time please also write in full

This has been changed this in the revised manuscript (line 73)

- ✓ In line 135, what does "these processes" mean?:

We will change this in the revised manuscript

- ✓ In line 176, "Chm has a much stronger effect on gene expression at 23{degree sign}C than at 25{degree sign}C" should be clearer with the context of starvation resilience. Otherwise please show direct differential analysis between the flies at two temperatures:

We will change this in the revised manuscript

- ✓ In 182, please explain more about the drug and how it is relevant to this study:

Ad libitum indicates unrestricted access to food. This has now been modified in the revised manuscript for better clarity

- ✓ In Figure 4, it's interesting to see the control flies at 23{degree sign}C could build up the trehalose. Would this be related to why control flies at 23{degree sign}C were the longest-lived group?:

We'd like to thank the reviewer for pointing this out. Indeed, trehalose serves as a buffer for carbohydrates in flies and may indeed be a natural response to alleviate the metabolic burden of starvation. The fact that trehalose is already high in chm mutants suggests that they indeed have an issue in regulating metabolism under fed conditions, which is then further enhanced during starvation. As trehalose has been shown to extend lifespan in C.elegans (Honda et al, 2010) it is tempting to speculate that the high concentration of trehalose may be one cause for the increased lifespan of chm mutant flies (Peleg et al, 2016) . We've discussed this hypothesis in the revised manuscript.,

- ✓ The numbers of Supplemental Figures 3-6 are wrong, please fix:

We have corrected this in the revised manuscript

Reviewer #2 (Minor comments):

Supplementary figure numbers should be corrected.

We will correct this in the revised manuscript

Reviewer #2 (Significance/General assessment):

- ✓ The authors don't give any information regarding the changes in the levels of various trehalose metabolism proteins being responsible for the higher trehalose levels in chm mutants, or speculate if it contributes to the starvation sensitivity of chm mutants at 23oC.

The expression data for enzymes and proteins involved in trehalose metabolism and their acetylation status in the control as well as the chm mutant flies are shown in Appendix S5-6 of the revised manuscript. The data are extracted from proteome and acetylome data published in our previous study (Venkatasubramani AV et al., 2023)

Reviewer #3 (Major comments):

- ✓ An illustration depicting a model of the underlying processes would be useful.

This is now included in the revised manuscript as panel G of the new figure 6.

Reviewer #3 (Minor comments):

- ✓ Overall, the manuscript is well-written and easy to follow. However, there are some confusing statements. For instance, the authors write, "which was even larger at lower temperatures" (lines 94-95). It should be clarified what exactly is larger.

We overhauled the entire manuscript, which led to a rephrasing and clarification of the sentence.

- ✓ Instead of using terms like "lighter," it would be more appropriate to use "lower weight" for scientific writing.

We corrected this in the revised manuscript

Reviewer #1 (Major comments):

- ✓ When using Chm RNAi line and Chm mutant line, at least some validation needs to be done to show the expression level of Chm, with e.g. qPCR.

These have already been validated with a Western blot in our previously published manuscript 2023 (Fig. EV1A (chm^{RNAi}) and EV1D ($chm^{MYST/+}$) (Venkatasubramani et al, 2023)).

Reviewer #3 (Major comments):

- ✓ The authors found that H4K12ac levels were significantly lower in $chm^{MYST/+}$ flies and that H4 di-acetylation was influenced by temperature. It would be beneficial to conduct ChIP-seq analysis of these marks to see if they correlate with the differentially expressed genes in the authors' RNA-Seq datasets.

We agree with the reviewer that this would have been very beneficial. Unfortunately, most H4ac antibodies show an unspecific activity towards poly-acetylated epitopes (Rothbart et al, 2012), which is particularly strong for the ones recognizing H4K12ac. Moreover, our overexpression data suggest that the changes in gene expression may not only be due to Chm's ability to acetylate H4.

- ✓ While the findings are interesting, the data do not provide insights into why an acetyltransferase might affect the observed phenotype. The authors attempt to discuss

this under the section titled "What Can Cause the Temperature-Specific Changes in Histone Acetylation Levels?" However, it would be helpful to include supporting experimental evidence.

We have removed this section now to improve the flow of the discussion, but as mentioned in the comments to reviewer 2 above we have now included measurements of acetyl-CoA precursor molecules at different temperatures and metabolic rescue experiments in the revised manuscript.

References

- Groef SD, Wilms T, Balmand S, Calevro F & Callaerts P (2021) Sexual Dimorphism in Metabolic Responses to Western Diet in *Drosophila melanogaster*. *Biomolecules* 12: 33
- Honda Y, Tanaka M & Honda S (2010) Trehalose extends longevity in the nematode *Caenorhabditis elegans*. *Aging Cell* 9: 558–569
- Jang T & Lee KP (2018) Context-dependent effects of temperature on starvation resistance in *Drosophila melanogaster*: Mechanisms and ecological implications. *J Insect Physiol* 110: 6–12
- Klepsatel P, Wildridge D & Gálíková M (2019) Temperature induces changes in *Drosophila* energy stores. *Sci Rep* 9: 5239
- Kubli E (2010) Sexual Behavior: Dietary Food Switch Induced by Sex. *Curr Biol* 20: R474–R476
- Lee KP & Jang T (2014) Exploring the nutritional basis of starvation resistance in *Drosophila melanogaster*. *Funct Ecol* 28: 1144–1155
- Peleg S, Feller C, Forne I, Schiller E, Sévin DC, Schauer T, Regnard C, Straub T, Prestel M, Klima C, et al (2016) Life span extension by targeting a link between metabolism and histone acetylation in *Drosophila*. *EMBO reports* 17: 455–469
- Rothbart SB, Lin S, Britton L-M, Krajewski K, Keogh M-C, Garcia BA & Strahl BD (2012) Poly-acetylated chromatin signatures are preferred epitopes for site-specific histone H4 acetyl antibodies. *Scientific Reports* 2: 489
- Venkatasubramani AV, Ichinose T, Kanno M, Forne I, Tanimoto H, Peleg S & Imhof A (2023) The fruit fly acetyltransferase *chameau* promotes starvation resilience at the expense of longevity. *EMBO Rep*
- Wat LW, Chao C, Bartlett R, Buchanan JL, Millington JW, Chih HJ, Chowdhury ZS, Biswas P, Huang V, Shin LJ, et al (2020) A role for triglyceride lipase *brummer* in the regulation of sex differences in *Drosophila* fat storage and breakdown. *PLoS Biol* 18: e3000595
- Wong R, Piper MDW, Wertheim B & Partridge L (2009) Quantification of Food Intake in *Drosophila*. *PLoS ONE* 4: e6063

October 13, 2025

RE: Life Science Alliance Manuscript #LSA-2025-03524-T

Axel Imhof
Univ. of Munich
Biomedical Center

Dear Dr. Imhof,

Thank you for transferring your manuscript entitled "Chameau (HBO1) regulates starvation resistance in *Drosophila melanogaster* in a temperature-dependent manner" to Life Science Alliance. As we discussed over email, please consider the following minor adjustments to the text in order to more directly align the conclusions presented with the observations made here.

Lines 138-9: Please adjust the claim on acetylation vs methylation as a metabolic sensor. The data presented so far do not fully exclude involvement of methylation and do not independently establish acetylation levels alone as a sensor of metabolic state.

Lines 280-2: The claim made on a chm-specific effect of acetyl-CoA supplementation is somewhat confusing given that the increase in survival was independent of genotype, as noted in the results and in subsequent lines 285-6. Please consider rephrasing this.

Discussion: Please include acknowledgement that 1) measured acetyl-CoA levels were inconsistent which, although perhaps not surprising, would have offered important support for the proposed model; and 2) that ChIP-seq or a similar assay could in theory provide greater mechanistic insight in a future study, including potential caveats on this method as appropriate.

We would be happy to publish your paper in Life Science Alliance pending the text changes noted above as well as final revisions necessary to meet our formatting guidelines.

- Please be sure that the authorship listing and order is correct.
- Please add a Category for your manuscript in our system.
- Please add Keywords for your manuscript in our system.
- Please add a Summary Blurb/Alternate Abstract in our system.
- Please add the X and Bluesky handles of your host institute/organization as well as your own or/and one of the authors in our system.
- Please add ORCID ID for secondary corresponding author--you should have received instructions on how to do so.
- The titles in both the system and the manuscript file must be consistent with each other.
- Please move all the main tables at the end of manuscript file or upload them separately in .docx or .xlsx file format.
- Please label tables consecutively.
- LSA allows supplementary figures, but no EV Figures; please update your labels and callouts for the Supplementary Figures (figure labels: Figure EV1=Figure S1; figure callouts in the manuscript Fig EV1A=Fig S1A); Same applies for tables: labels and callouts should be adjusted to Supplementary Table 1 etc.
- Please adjust the labels for Appendix figures same as for EV Figures. All the figures that are not main figures must follow the labeling for Supplementary Figures (Figure S1, S2 etc.).
- Please add your main, supplementary figure, and table legends to the main manuscript text after the references section (captions should be contained within figures).
- Please rename the datasets as supplementary tables - both in their titles and in their callouts in the manuscript text.
- Please add a callout for Figure 1B to your main manuscript text.
- Please present figure panels in consecutive order in the captions (Figure 1 caption needs to be rectified).

To upload the final version of your manuscript, please log in to your account: <https://lsa.msubmit.net/cgi-bin/main.plex>
You will be guided to complete the submission of your revised manuscript and to fill in all necessary information. Please get in

touch in case you do not know or remember your login name.

A. FINAL FILES:

B. MANUSCRIPT ORGANIZATION AND FORMATTING:

Thank you for your attention to these final processing requirements. Please revise and format the manuscript and upload materials as soon as you are able.

Sincerely,

October 23, 2025

RE: Life Science Alliance Manuscript #LSA-2025-03524-TR

Dr. Axel Imhof
Ludwig-Maximilians-Universität München
Biomedical Center
Großhadernerstr. 9
Planegg-Martinsried, Bayern 82152
Germany

Dear Dr. Imhof,

Thank you for submitting your Research Article entitled "Chameau (HBO1) regulates starvation resistance in *Drosophila melanogaster* in a temperature-dependent manner", and for your attention to the minor points we detailed in our previous letter. It is a pleasure to let you know that your manuscript is now accepted for publication in Life Science Alliance. Congratulations on this interesting work!

DISTRIBUTION OF MATERIALS:

Again, congratulations on a very nice paper. I hope you found the review process to be constructive and are pleased with how the manuscript was handled editorially. We look forward to future exciting submissions from your lab.

Sincerely,
